



**Characterising optical array particle imaging probes: implications**
**for small ice crystal observations**
**Sebastian O'Shea[1], Jonathan Crosier[1,2], James Dorsey[1,2], Louis Gallagher[3], Waldemar**
**Schledewitz[1], Keith Bower[1], Oliver Schlenczek[4,5,a], Stephan Borrmann[4,5] Richard Cotton[6],**
**Christopher Westbrook[7], and Zbigniew Ulanowski[1,8,9]**
[1] {School of Earth and Environmental Sciences, University of Manchester, UK}
[2] {National Centre for Atmospheric Science, University of Manchester, UK}
[3] {Department of Physics and Astronomy, University of Manchester, UK}
[4] {Particle Chemistry Department, Max Planck Institute for Chemistry, Germany}
[5] {Institute for Atmospheric Physics, Johannes Gutenberg University, Germany}
[6] {Met Office, Exeter, UK}
[7] {Department of Meteorology, University of Reading, UK}
[8] {Centre for Atmospheric and Climate Processes Research, University of Hertfordshire, UK}
[9] {British Antarctic Survey, NERC, Cambridge, UK}
[a] {now at: Max Planck Institute for Dynamics and Self-Organization, Göttingen, Germany}.
Correspondence to: j.crosier (j.crosier@manchester.ac.uk)
**Abstract**
The cloud particle concentration, size and shape data from optical array probes (OAPs) are
routinely used to parameterise cloud properties and constrain remote sensing retrievals. This
paper characterises the optical response of OAPs using a combination of modelling, laboratory
and field experiments. Significant uncertainties are found to exist with such probes for ice
crystal measurements. We describe and test two independent methods to constrain a probe's



sample volume that removes the most severely mis-sized particles: (1) greyscale image analysis
and (2) co-location using stereoscopic imaging. These methods are tested using field
measurements from three research flights in cirrus. For these cases, the new methodologies
significantly improve agreement with a holographic imaging probe compared to conventional
data processing protocols, either removing or significantly reducing the concentration of small
ice crystals ($<200\,\mu m$) in certain conditions. This work suggests that the observational evidence
for a ubiquitous mode of small ice particles in ice clouds is likely due to a systematic instrument
bias. Size distribution parameterisations based on OAP measurements need to be revisited using
these improved methodologies.

## 11    1    Introduction

A significant amount of our current understanding of cloud microphysics is based on in-situ
measurements made using Optical Array Probes (OAPs). This includes how cloud properties
are parameterised in numerical climate/weather models and how they are retrieved from remote
sensing datasets, including global cloud and precipitation monitoring satellites such as NASA's
GPM (Global Precipitation Mission), CloudSat and CALIPSO (Cloud-Aerosol Lidar and
Infrared Pathfinder Satellite Observation) (Mitchell et al., 2018; Sourdeval et al., 2018; Ekelund
et al., 2020; Eriksson et al., 2020; Fontaine et al., 2020).
Optical array probes are a family of instruments that have been widely used by the cloud physics
community for the last 40+ years. Primarily OAPs have been operated on research aircraft
(Wendisch and Brenguier, 2013). They collect images of cloud particles and are used to derive
cloud particle concentration, size and crystal habit (shape). Optical array probes operate by
recording a shadow image as a particle crosses a laser beam that is illuminating a 1-D linear
array of photodiode detectors. If the light intensity at any of the detectors drops below a
threshold value, the probe records an image of the particle and the corresponding timestamp.
A two-dimensional image of the particle is constructed by appending consecutive one
dimensional "slices" from the array of detectors as the particle moves perpendicular to the laser
beam due to the motion of air through the probe.
The rate at which data needs to be acquired from the detectors depends on the air speed through
the probe and the required image resolution. For example, when operated on research aircraft
at a typical airspeed of $100\ m\ s^{-1}$ image slices from the detectors are acquired every $0.1\ \mu s$ to
achieve an image resolution of $10\ \mu m$. Modern OAPs have 64 to 128 element detector arrays





with pixel resolutions ranging from 10 to 200 µm. Monoscale probes use a 50% drop in intensity
as a threshold for detection which results in 1-bit binary images (Knollenberg, 1970; Lawson
et al., 2006), while most greyscale OAPs have three intensity thresholds, which result in 2-bit
grayscale images (Baumgardner et al., 2001).
When particles pass through the object plane of a probe they are in focus and accurate digitized
images are recorded. When particles are offset from this plane the diffraction of light by the
particle alters the size and shape of the recorded image from its original form. When the distance
from the object plane (Z) is sufficiently large the reduction in light intensity at the detector will
no longer exceed the detection threshold. This distance is known as the probe's depth of field
(DoF). For large particle sizes the DoF is constrained by the physical separation between the
laser transmit and receive optics, which are in protruding structures referred to as "arms". The
following equation is generally used to define the DoF of monoscale probes using a 50%
intensity threshold for detection (Knollenberg, 1970)

$$DoF = \pm \frac{cD_0{}^2}{4\lambda}$$

15                                                                                    Equation 1

where $D_0$ is the particle diameter and $\lambda$ is the laser wavelength. c is a dimensionless constant,
typically between 3 and 8 (Lawson et al., 2006; Gurganus & Lawson, 2018). The DoF is used
to determine particle concentration by dividing the number of counts by the sample volume
(SVol), which is given by,

$$SVol = TAS \int_{-DoF}^{+DoF} \left( ER - D(Z) \right) dZ,$$

21                                                                                    Equation 2

Where TAS is the true air speed, E is the number of detector array elements, R is pixel size of
the probe and D is the image diameter. The integration of the effective array width (ER-D(Z))
is performed over whichever is smaller out of the DoF and the armwidth of the probe.
For spherical particles, corrections exist which correct for the diffraction effects of sampling
offset from the object plane, which allows the calculation of the true particle size from the
measured image size. Korolev et al. (1991) show that the diffraction from spherical liquid drops
can be approximated by the Fresnel diffraction from an opaque disk. The ratio of the measured
image diameter to the true particle diameter is a function of the dimensionless distance from
the object plane $Z_d$:





$$Z_d = \frac{4\lambda Z}{D_0{}^2}$$

2                                                                                     Equation 3

Korolev et al. (2007, hereafter K07) describes how the size of the bright spot at the centre of a
diffraction image can be used to determine a sphere's distance from the object plane and
therefore true size. O'Shea et al. (2019, hereafter O19) show that this correction is effective for
modern OAPs up to approximately $Z_d = 6$, after which the images are too fragmented to reliably
correct. O19 show that greyscale information can be used to remove these fragments by
identifying the distance from the object plane of spherical particles in the range $Z_d = 3.5$ to $8.5$.
This allows a new DoF to be defined that excludes the fragmented images.
There has been significant discussion in the literature about the presence of high concentrations
of small ice particles (< 200 µm) observed by OAPs in cirrus and other types of ice clouds
(Jensen et al., 2009; Korolev et al., 2011). O19 shows that fragmented images near the edge of
the DoF have the potential to significantly bias OAP particle size distributions (PSDs) and result
in an artificially high concentration of small particles.
This paper quantifies the uncertainties in OAP size and shape measurements of non-spherical
ice crystals and presents corrections that removes large biases from OAP datasets. In Sect. 3.1,
3-D ice crystal analogs are repetitively passed through the sample volume of an OAP at
different distances from the object plane. These results are used to examine the ability of a
diffraction model based on angular spectrum theory to characterise the response of OAPs. In
Sections 3.2 to 3.5 a variety of ice crystals from commonly occurring habits are tested with the
diffraction model to quantify how OAP image quality degrades throughout a probe's sample
volume. Section 4 suggests and tests methods to improve OAP data quality. The impact these
results have on ice crystal PSDs is examined using field measurements collected during three
research flights in frontal cirrus. The impacts OAP measurement bias has on our understanding
of ice cloud microphysics are discussed in Sect. 5.





**2    Methods**
**2.1    Optical array probes**
This paper uses data from two types of commercially available OAP: a CIP-15 (Cloud imaging
probe, DMT Inc., USA; Baumgardner et al., 2001) and a 2D-S (2D stereo, SPEC Inc., USA;
Lawson et al., 2006). The CIP-15 has a 64 element photodiode array and effective pixel size of
15 µm. The laboratory experiments were conducted with a CIP-15 with an arm separation of
70 mm (Sect. 3.1) and the field measurements with a second CIP-15 with an arm separation of
40 mm (Sect. 4.1). Images are recorded at three greyscale intensity thresholds. For this work
they were set to the manufacturer default settings of 25%, 50% and 75%. The 2D-S consists of
two optical arrays and lasers orientated at right angles to each other and the direction of motion
of the particles/aircraft. The laser beams overlap at the centre of the probe's arms, and each pair
of transmit/receive arms are separated by 63 mm. Each optical array has 128 elements and 10
µm pixel resolution. The 2D-S is a monoscale probe with a single 50% intensity detection
threshold. Both probes are fitted with anti-shatter tips to minimise ice shattering on the leading
edge of the probe during field measurements. This was further minimised by removing particles
with inter-arrival times less than $1 \times 10^{-5}$ s when calculating PSDs from field measurements
(Field et al., 2006).
Baumgardner & Korolev (1997) show that the electronic time response of older probes can
significantly reduce the DoF of small particles. This affect has been minimised in more modern
probes such as the 2D-S and CIP-15, which have an order of magnitude faster time response.
A range of definitions have been used to define the diameter of ice crystals from OAP images.
Here we test three metrics that have been widely used by the community. First, the mean of the
particle extent along the axes parallel and perpendicular to the optical array (mean X-Y
diameter). Second, the diameter calculated using $D = (4A/\pi)^{1/2}$ where A is the particle area
calculated as the sum of the pixels (circle equivalent diameter). Third the major axis length of
the ellipse that has the same normalized second central moments as the region (maximum
diameter).
An image frame from and OAP may contain more than one object, where individual objects are
defined as collections of pixels with 8-neighbor connectivity. This can be due to diffraction,
with a single particle appearing as more than one object as the structure and intensity of the
transmitted light degrades due to poor focus. However, it may also be due to shattering causing





multiple particles to have sufficiently small separations that they are captured in the same image
frame or occasionally when there are very high concentrations of ambient particles. A particle
sizing metric can either relate to the largest object in an image frame or use the bounding box
encompassing all objects. Some previous studies have filled any internal voids within objects
in an image frame. For this work, unless otherwise stated both the mean X-Y, maximum and
circle equivalent diameters are calculated using the bounding box encompassing all objects in
an image frame and any internal voids are not filled.

## 2.2   Ice crystal analogs

Three-dimensional ice crystal analogs were grown from a sodium fluorosilicate solution
(Ulanowski et al., 2003). These analogs have similar crystal habits to ice and a refractive index
of 1.31, virtually identical to that for ice at visible wavelengths. Three rosette shapes were used
in these experiments with approximate diameters 118 µm (ROS118), 250 µm (ROS250) and
300 µm (ROS300) (Fig. 1). The CIP-15 was mounted as shown in Fig. 2 so that the laser beam
was vertically aligned. Each analog was in turn placed on an anti-reflective optical window that
was attached to a 3-axis translation system that allowed the analog's 3D position to be
controlled. The stages that moved along the axes parallel (X axis) to the diode array and laser
beam (Z axis) each had a unidirectional position accuracy of 15 µm and travel ranges of 100
mm and 150 mm, respectively (X-LRM050A and Z-LRM150A, Zaber Technologies Inc,
Canada). Movements along the axis that air flows through the probe under normal operation (Y
axis) were made using a belt-driven stage with a maximum speed of 1.1 m s$^{-1}$, positional
accuracy of 200 µm and maximum travel range of 70 mm (X-BLQ0070-EO1, Zaber
Technologies Inc, Canada).
CIP-15 images of the ice crystal analogs were collected by moving them through the laser beam
along the axis of airflow. For each analog this was repeated 5 times before its position was
stepped in 0.5 mm increments between the probe's vertical arms (along the Z axis). This allows
images of the analogs to be compared at different distances from the object plane.
Images were post-processed to take account of any difference in velocity between the stage and
the CIP-15 data acquisition rate by resampling the images along the axis perpendicular to the
optical array. This was performed to match the aspect ratio at Z=0 of the CIP-15 image and a





microscope image of each analog. This typically corresponded to a particle stage velocity of ~
$0.1 \text{ m s}^{-1}$.

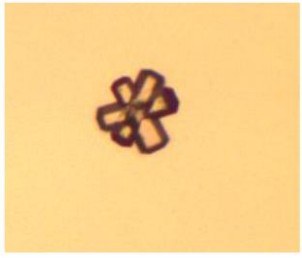 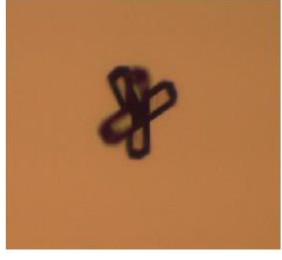 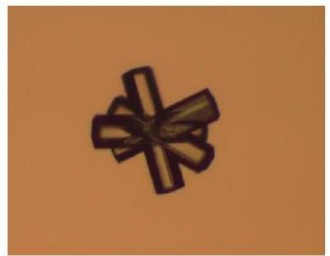

*Figure 1. Microscope images of sodium fluorosilicate crystals that were used as analogs for*
*ice crystals. These are referred to as ROS118 (left), ROS250 (middle) and ROS300 (right).*

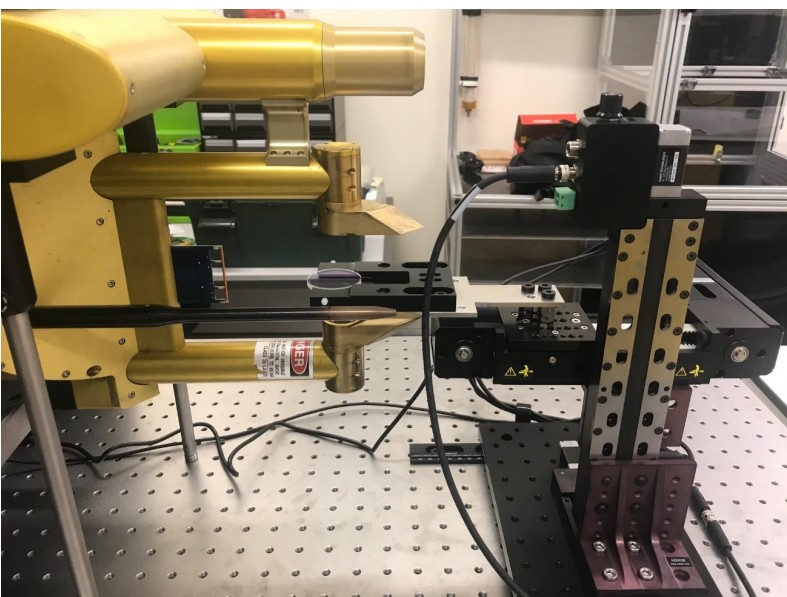



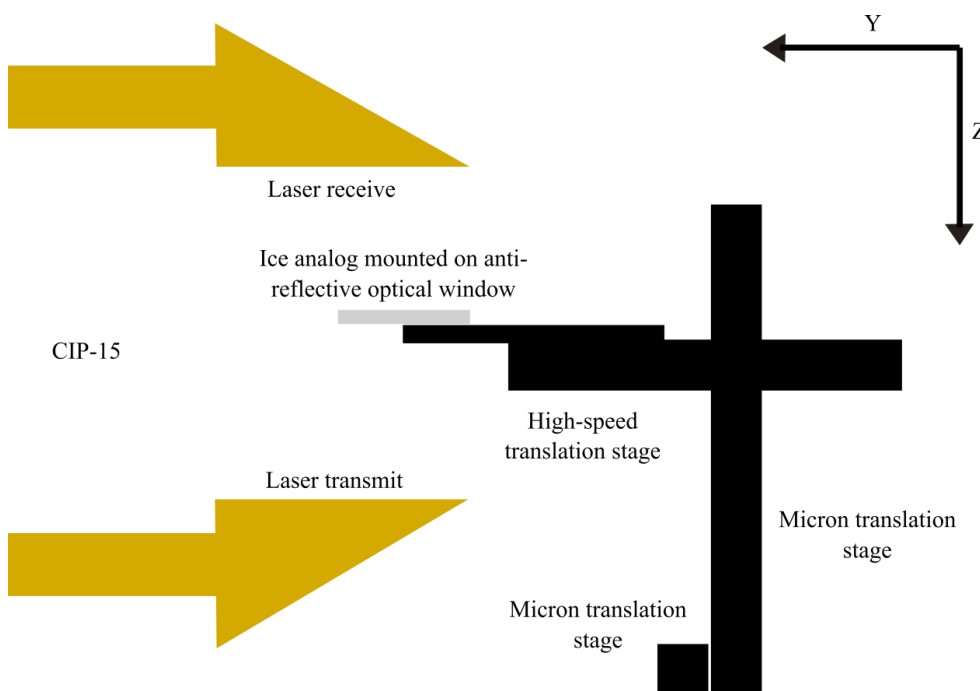

*Figure 2. Top panel. Image of the experimental setup for the ice crystal analog tests of the CIP-*
*15. The bottom panel shows a schematic of the experimental set-up. The CIP-15 is horizontally*
*mounted on the left of the image. The translation stages used to move ice crystal analogs*
*through CIP-15 sample volume is shown on the right of the image. The X axes is perpendicular*
*to the plane of drawing.*
**2.3   Synthetic data (Angular spectrum theory)**
Theoretical shadow images of 2D non-spherical shapes were calculated using a diffraction
model based on Angular Spectrum Theory (referred to as the AST model). Several previous
studies describe this model in detail (Vaillant de Guélis et al., 2019a, 2019b). We initialised the
model using a 2-D binary image of an opaque shape at the object plane (Z = 0) and calculate
the wave field for different positions between the probe arms in the Z axis. This model has been
shown to give good agreement with OAP images of several types of 2D rectangular columns
using images printed on a rotating disk (Vaillant de Guélis et al., 2019a).





In this study, we use a variety of different shapes to initialise the model. In Sect. 3.1, the
diffraction model is compared to CIP15 images of 3D ice crystal analogs. To initialise the model
for the comparisons with ROS250 and ROS300 the CIP-15 image of them at Z=0 is used. Due
to the smaller size of ROS118 and coarse pixel size of the CIP-15, a microscope image of the
analog is used to initialize the model. This image converted to a binary image.
In Sect. 3.2 the quality of OAP images of commonly occurring ice crystal habits is explored.
This is done by initialising the model with a variety of different ice crystal images. The ice
crystal dataset contains 1060 images that were collected using a Cloud Particle Imager (CPI,
SPEC Inc., USA) and has previously been used to train habit recognition algorithms
(Lindqvist et al., 2012; O'Shea et al., 2016). It includes images of ice crystals from arctic,
mid-latitude and tropical clouds. These images have been manually classified into 7 habits
(rosette, column/bullet, plate, quasi-spherical, column-aggregate, rosette-aggregate and
plate-aggregate). To initialise the model each CPI image was converted to a binary image.
Shadow images were calculated every 2 mm for the range Z = 0 to 100 mm. These images were
averaged to 10 µm pixel resolution, which is typical of modern OAPs. All simulations were
performed using a light wavelength of 0.658 µm.
An example simulation for a rosette crystal is shown in Fig. 3 and a column in Fig. 4, the top
left panels show the images at Z=0 that are used to initialise the model. The other panels show
images of the crystals at different distances from the object plane. Green, blue and black pixels
correspond to decreases in detector intensity of 25 to 50%, 50 to 75% and > 75%, respectively.
Figures 3 and 4 show the rapid deterioration in image quality within a few mm of the object
plane, which will impact derived properties such as particle size, number and habit. This
compares to many 10s of mm for the typical arm separation of modern OAPs.

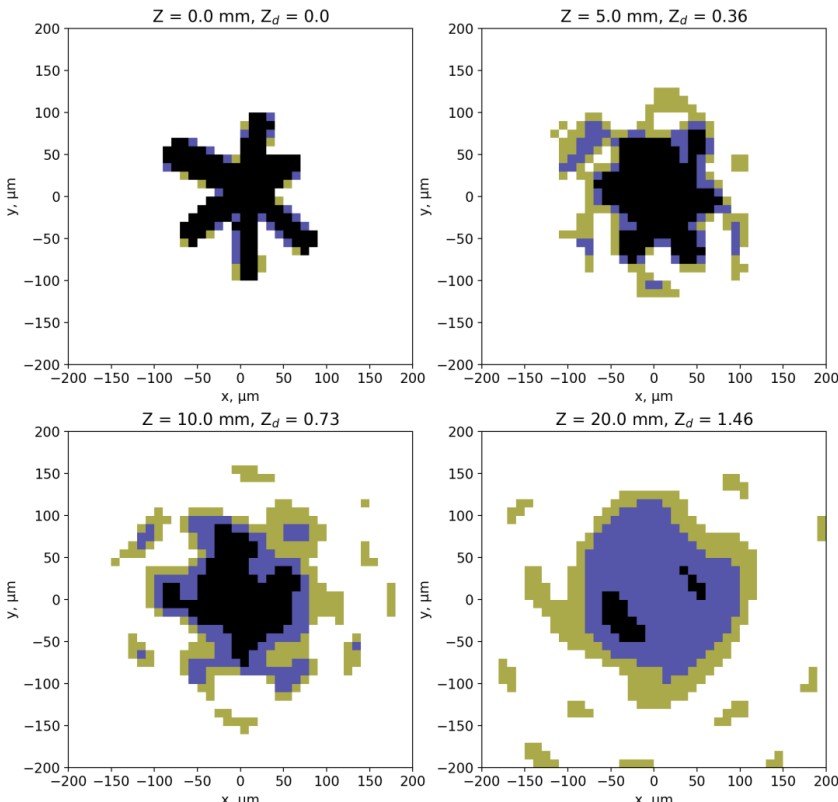

*Figure 3. Diffraction simulations from an image of a rosette crystal collected in cirrus cloud*
*using a CPI (see text for details). Top left panel show the image at Z=0 that is used to initialise*
*the model. The other panels show images at different distances from the object plane (Z= 5, 10*
*and 20 mm). Green, blue and black pixels correspond to decreases in detector intensity of 25*
*to 50%, 50 to 75% and greater than 75%, respectively.*

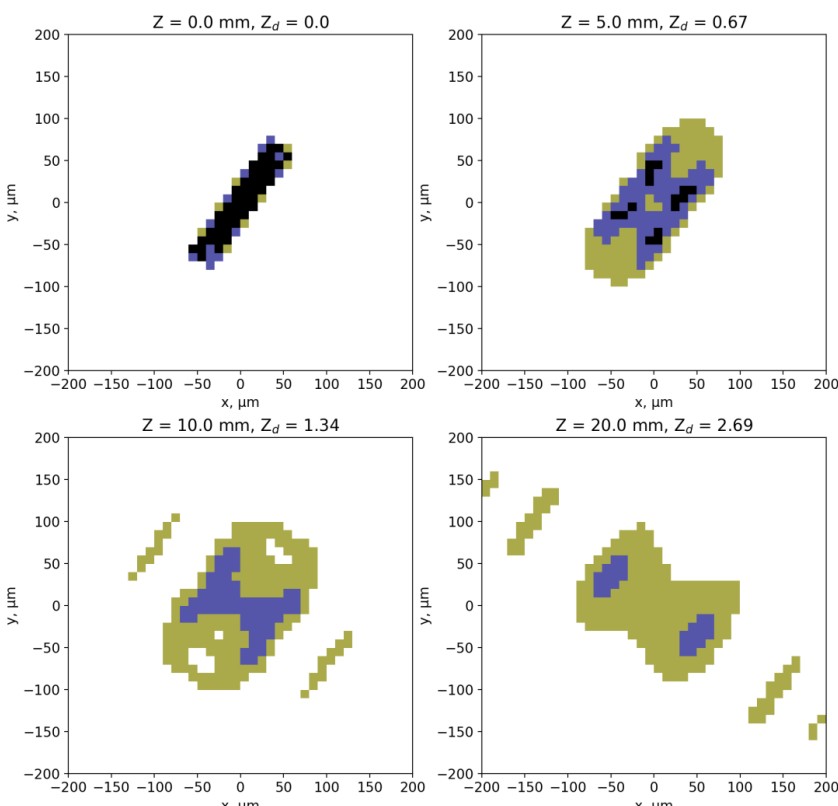

*Figure 4. Same as Figure 3 but for a column.*

**2.4 Aircraft measurements**
This paper uses measurements from three flights by the FAAM Bae-146 research aircraft
sampling frontal cirrus in the UK on the 11 March 2015 (nominal flight number B895), 7
February 2018 (C078) and 23 April 2018 (C097). The first two flights have previously been
described in detail in by O'Shea et al. (2016) and O19. For all 3 flights the aircraft performed
straight and level runs of approximately 10 minutes at different temperatures within the cloud.
Ice crystals were dominated by rosettes, columns and aggregates. Data from a 2D-S is available
for the 11 March 2015 and CIP-15 for 7 February 2018 and 23 April 2018. On all flights the





FAAM BAe-146 was fitted with a holographic imaging probe (HALOHolo). HALOHolo has a
$6576 \times 4384$ pixel CCD detector with an effective pixel size of 2.95 μm and arm separation of
155 mm. The probe acquires 6 frames per second, which equates to a volume sample rate of
~230 cm$^3$ s$^{-1}$. The detection of small particles is limited by noise in the background image.
Therefore, a minimum size threshold of 35 μm is applied, above which it is estimated that the
probe's detection rate is greater than 90% (Schlenczek, 2017). Shattered particles were
minimised by removing all particles with inter particle distances less than 10 mm (Fugal &
Shaw, 2009; O'Shea et al., 2016).
Section 4.2 shows a comparison between the 2D-S and a Cloud droplet probe (CDP, DMT Inc.)
during a flight in liquid stratus on 17 August 2018 (C031). The CDP sizes particles (3 to 50
μm) using the scattered light intensity assuming Mie-scattering theory and spherical particles
(Lance et al., 2010). The probe was calibrated during the campaign using glass spheres.

## 14    3    Results

### 15    3.1    OAP and AST model comparison using ice crystal analogs

This section compares CIP-15 images of ice crystal analogs with diffraction simulations using
the AST model. Figures 5-7 show the image size of the ice crystal analogs ROS118, ROS250
and ROS300 at different distances (Z) from the object plane measured by the CIP-15 (black
markers) and modelled using angular spectrum theory (red lines). Top left panels show the
image diameter (mean X-Y), while the particle area is shown in the top right, both use a 50%
drop in light intensity for the detection threshold. The other panels show different combinations
of simple greyscale ratios. The abbreviations $A_{25-50}$, $A_{50-75}$ and $A_{75-100}$ are used to denote the
number of pixels associated with a decrease in detector signal of 25 – 50%, 50 – 75% and 75 –
100%, respectively. Example CIP-15 images of the ice crystal analog ROS300 at 3 distance
from the object plane are shown in Fig. 8.
All three analogs have a general trend of diameter initially increasing with Z. The full DoF was
sampled for ROS118 and shows the images fragmenting and diameter decreasing near the edge
of the DoF. In addition to these general trends there is a significant amount of fine scale
structure that is specific to each sample. There is a general trend of the greyscale ratio $A_{75-100}$
decreasing with Z, while both $A_{25-50}$ and $A_{50-75}$ initially increase for all 3 analogs. Like the





diameter vs Z plots there is a significant amount of fine scale structure overlaying these general
trends.
In general, the AST model can capture the large-scale structure in these measured parameters,
although some discrepancies are present in the finer detail. For ROS118 the DoF from the
experiments and the model agree to within ±1 mm (Fig. 5). The size and greyscale parameters
calculated from CIP-15 images are not completely symmetrical about Z=0. The reason for this
is unclear, potential causes are if the CIP-15 laser beam is not perfectly collimated, additional
refraction caused by the optical window used to mount the sample, or changes to the CIP-15
background/dark current calculation due to attenuation by the optical window.

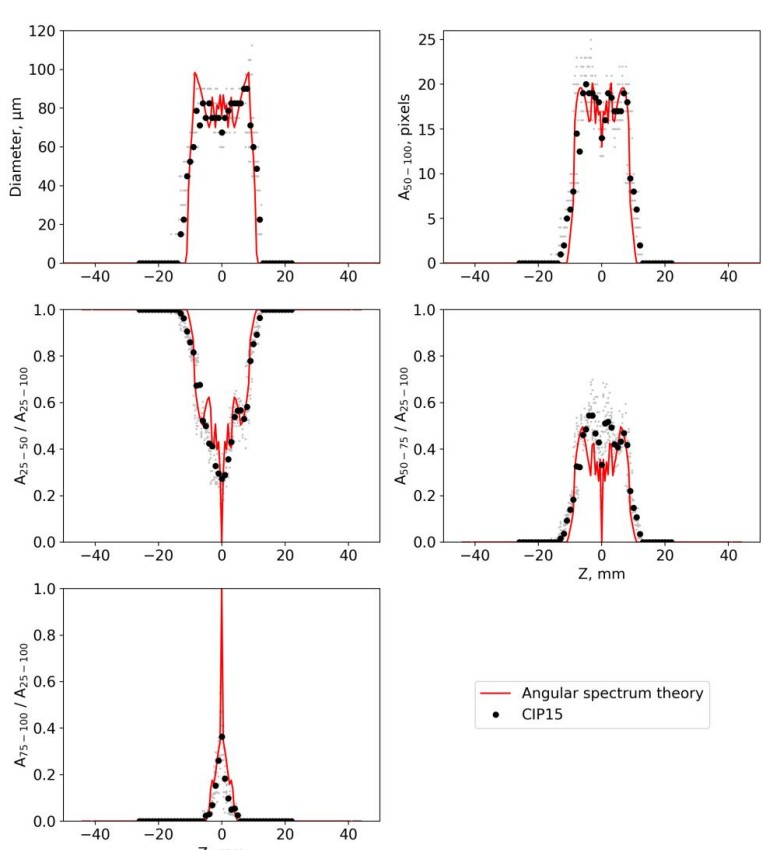




*Figure 5 A comparison between CIP-15 images and diffraction simulations (red lines) of the*
*ice crystal analog ROS118. Grey dots show data from individual CIP-15 images and black dots*
*show the median for each 1 mm Z bin. Top left panel shows the mean X-Y image diameter. Top*
*right shows the number of pixels using 50% detection thresholds. Other panels show the ratio*
*of number of pixels (area) at different greyscale thresholds.*

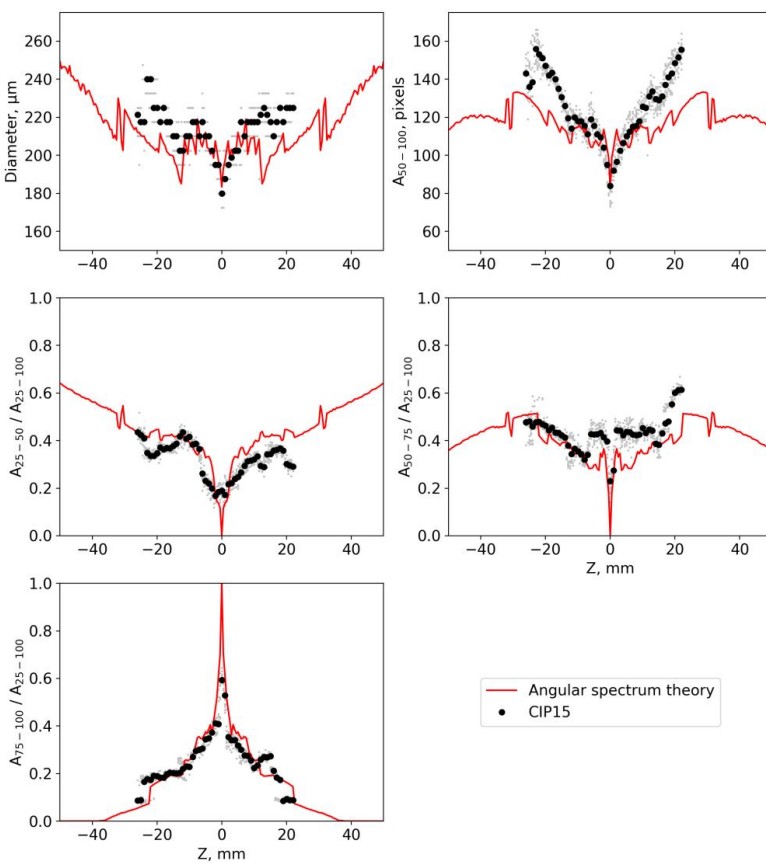

*Figure 6. Same as Fig. 5 but for the ice crystal analog ROS250.*



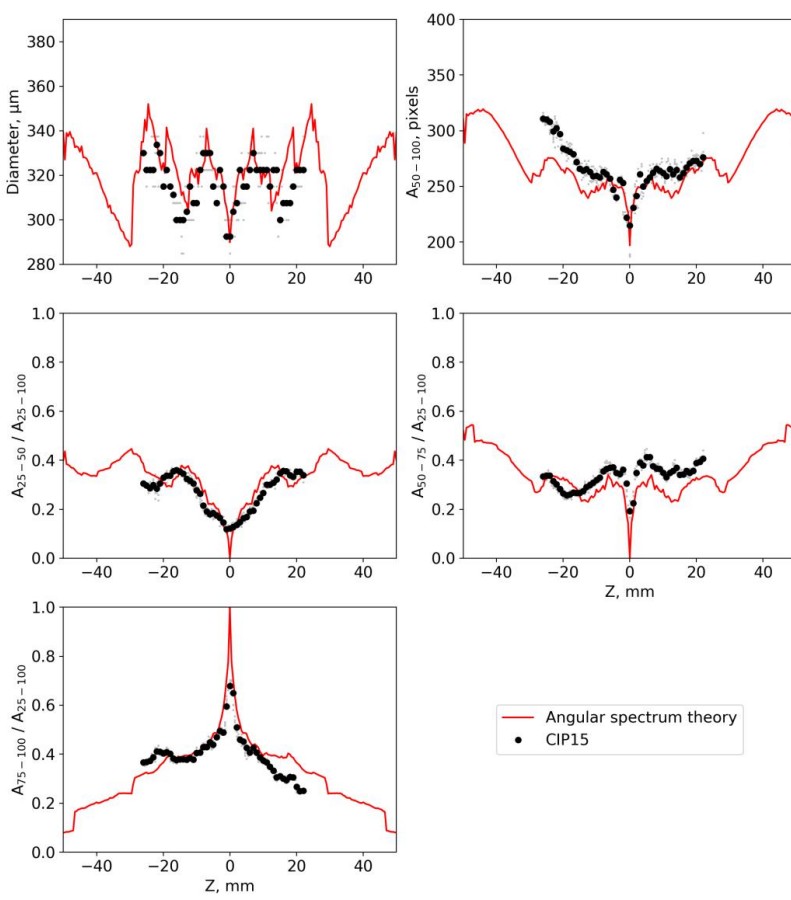

3    *Figure 7. Same as Fig. 5 but for the ice crystal analog ROS300.*





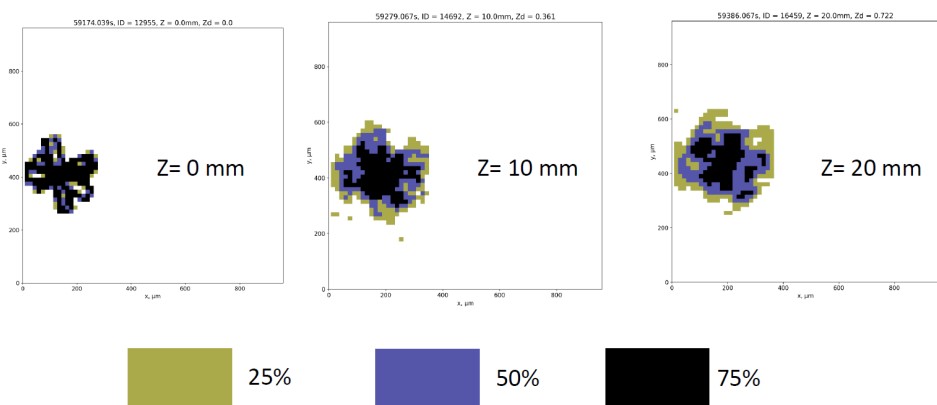

*Figure 8. CIP-15 images of the ice crystal analog ROS300 at 3 distances from the object plane.*
**3.2    OAP ice crystal sizing**
Having investigated the performance of the AST model using 3-D analogs of complex ice, we
will now use the AST model to examine the ability of OAPs to correctly determine the size of
commonly occurring ice crystals. Figure 9 left panels show the ratio of the measured diameter
(D) to the true diameter ($D_0$) vs $Z_d$ for diffraction simulations of 1060 ice crystals. The data for
each individual ice crystal is shown as grey lines, while the coloured lines are the median for
each habit. Top panels show plots using the circle equivalent diameter, while the middle panels
use the mean X-Y diameter and maximum diameter. Right panels show histograms of $D/D_0$ for
each habit calculated for the $Z_d$ range from 0 to 10.
Figure 9 shows large differences in these relationships depending on whether the mean X-Y,
maximum or circle equivalent diameter is used to define the particle size. For the 1060 ice
crystal images used in this study the median $D/D_0$ over the $Z_d$ range from 0 to 8 is 1.1 using
circle equivalent diameter, 1.0 using the mean X-Y diameter and 1.0 using the maximum
diameter. However, there is significantly less variability between crystals using circle
equivalent diameter, which has an inter-quartile range $D/D_0$ of 0.2 compared to 1.1 and 1.3
using the mean X-Y and maximum diameters, respectively. This is also shown in Table S1-S3,
which gives the median and inter-quartile range $D/D_0$ at selected $Z_d$ for each habit using the
three different size metrics.



There is a general trend of increasing size with distance from the object plane. Over-sizing is
up to approximately 100%, 200% and 50% using mean X-Y, maximum and circle equivalent
diameters, respectively. However, the degree of oversizing is dependent on habit, with quasi-
spherical and plate aggregates most significantly over-sized using all D definitions. In
agreement with O19, once D reaches a maximum, further increases in Z cause the images to
fragment and their size to decrease until they are no longer visible.
K07 uses the size of the internal voids within images of droplets to determine their $Z_d$ and
correct their size. O19 shows that this algorithm is effective using modern OAPs for droplets
with $Z_d <$ ~6. For $Z_d > 6$ the images are too fragemented for their size to be corrected. The K07
approach was derived by considering Fresnel diffraction from opaque discs, and has only been
tested for images of spherical droplets. However, previous studies have applied K07 to images
of ice crystals (e.g. Davis et al., 2010). To examine the efficacy of this approach, Fig. 9 bottom
panels shows the mean X-Y diameter of the simulated images of ice crystals once K07 has been
applied. The ratio of their K07 corrected diameter to their true particle diameter is shown as a
function of $Z_d$ (left panel), while probability density functions of $D/D_0$ for each habit are shown
in the right panel. The median $D/D_0$ for the $Z_d$ range 0 to 8 is 0.9 and the inter-quartile range is
1.1. For a number of habits (rosette, plate, quasi-spherical, rosette-aggregate and plate-
aggregate) K07 reduces the number of over-sized particles across most of the DoF. For bullets,
columns and column aggregates K07 has minimal impact on the probe sizing. Like for droplets,
K07 is not able to remove the small image fragments that occur when a particle is near the edge
of the DoF.



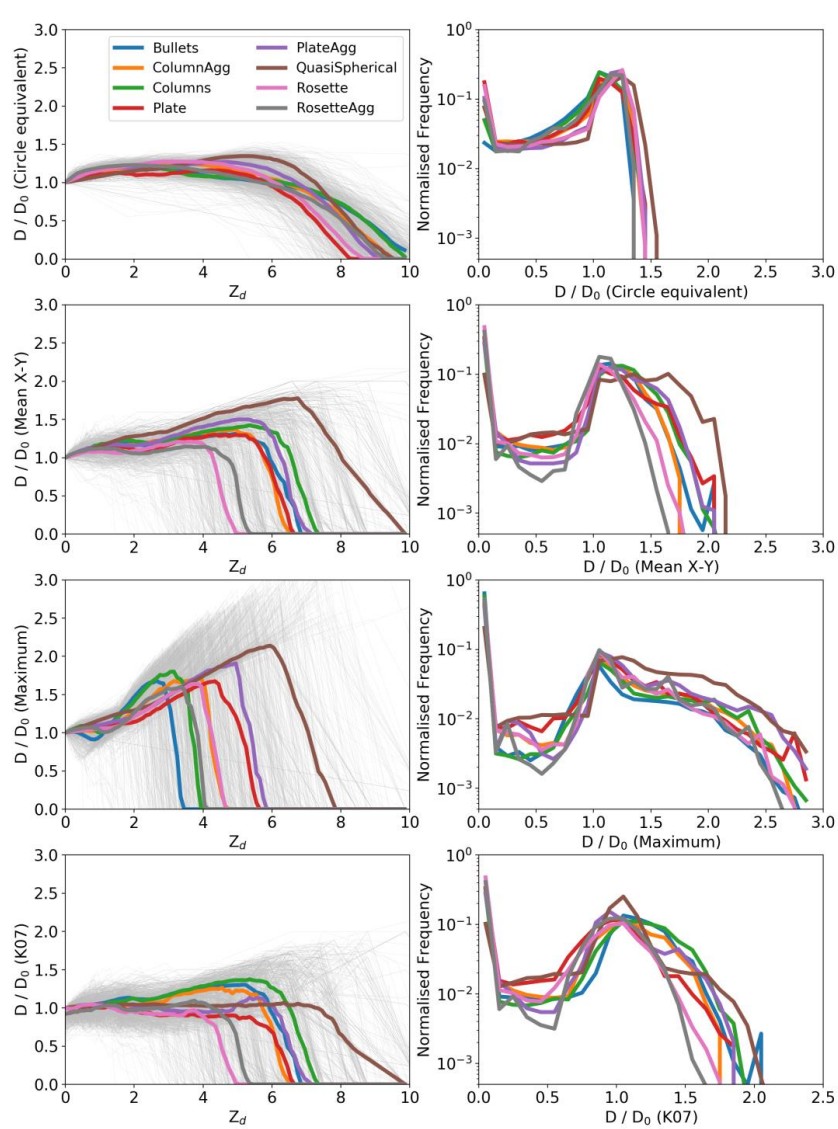



*Figure 9. Left panels show the ratio of the measured diameter (D) to the true diameter ($D_0$) vs*
$Z_d$ *for diffraction simulations of 1060 ice crystals. The data for each individual ice crystal is*
*shown as grey lines, while the coloured lines are the median for each habit. Right panels show*
*histograms of D/D0 for each habit calculated for the $Z_d$ range 0 to 10. Top panels show plots*
*using the circle equivalent diameter, while the middle panels use the mean X-Y and maximum*
*diameters. Bottom panels show the diameter corrected using K07.*

## 3.3    Depth of field dependence on particle habit

Uncertainty of derived physical quantities (e.g. number concentration) from OAPs is dependent
on the sample volume and therefore uncertainty in the DoF (see Eq. 2). The DoF of an OAP is
commonly calculated using Eq. 1 with a single c value. The variable c in this equation is the $Z_d$
where a particle is no longer detected by the OAP.  If a single c value is used this would need
to be independent of particle shape. Table 1 shows the median and inter-quartile range $Z_d$ where
particles are no longer visible for each habit using the maximum, mean X-Y and circle
equivalent diameters. Using mean X-Y the habit median DoF varies between $Z_d = 5.0$ and 9.9
for rosettes and quasi-spherical particles, respectively. Using the maximum as the particle
sizing metric the median DoF varies by a similar amount ranging between $Z_d = 3.4$ to 7.8 for
bullets and quasi-spherical crystals. In addition, particles have significant intra-habit variability
using both maximum and mean X-Y, with most habits DoF inter-quartile ranges greater than 2
$Z_d$. The variability is lower using circle equivalent diameter, with median DoFs ranging
between 8.2 and 10.2 for plates and bullets, respectively with habit inter-quartile ranges near 1
$Z_d$. As a result, derived physical quantities such as number concentration will have lower
uncertainty if circle equivalent diameter is used to define the particle size compared to
maximum and mean X-Y diameter.

|  |  | Bullets | Column-aggregates | Columns | Plates | Plate-aggregates | Quasi-spherical | Rosettes | Rosette-aggregates |
|---|---|---|---|---|---|---|---|---|---|
| Maximum | Median | 3.4 | 4.6 | 3.9 | 5.6 | 5.8 | 7.8 | 4.6 | 4.1 |
|  | IQR | 1.3 | 1.5 | 2.0 | 2.8 | 1.9 | 2.0 | 1.9 | 1.9 |
|  | Median | 6.8 | 6.6 | 7.2 | 7.0 | 7.0 | 9.9 | 5.0 | 5.4 |





| Mean X-Y | IQR | 2.0 | 1.7 | 2.1 | 3.0 | 2.4 | 2.0 | 2.0 | 2.0 |
|---|---|---|---|---|---|---|---|---|---|
| Circle equivalent diameter | Median | 10.2 | 9.4 | 9.9 | 8.2 | 9.0 | 9.4 | 8.6 | 9.2 |
| | IQR | 1.0 | 1.0 | 0.9 | 1.2 | 1.1 | 1.4 | 1.3 | 1.1 |

*Table 1. Median and inter-quartile range (IQR) normalised dimensionless distance from the*
*object plane ($Z_d$) where particles are no longer visible for different habits, this is equivalent to*
*c in Eq. 1*
**3.4   Greyscale information**
Greyscale information in OAP imagery has previously been used to filter severely mis-sized
images and enforce a DoF threshold that improves data quality (O19). Figure 10 shows
combinations of simple greyscale ratios as a function of $Z_d$ for the simulation of 1060 ice crystal
images described in the previous section. Left panels use the size metric mean X-Y diameter in
the $Z_d$ calculation, whereas the right panels use circle equivalent diameter in $Z_d$ calculation.
Like the ratio $D/D_0$ (Fig. 9), the greyscale ratios also show significant variability between habits
as a function of $Z_d$. Figure 10 shows this variability is greater if mean X-Y diameter is used to
calculate $Z_d$, though it is still significant using circle equivalent diameter. The variability is
larger still using maximum diameter (not shown).
O19 uses simple greyscale ratios to determine $Z_d$ for spherical liquid droplets near the edge of
the DoF ($3.5 < Z_d < 8.5$). This allows a new DoF to be defined that excludes fragmented images,
removing significant biases in the PSD. This is possible since all spherical droplets independent
of size have the same greyscale ratios at a given $Z_d$. Figure 10 shows that this is not true for ice
crystals where the initial shape of the ice crystal has an impact on the greyscale ratios at a given
$Z_d$. As a result, O19 cannot be used to determine $Z_d$ in the same way.



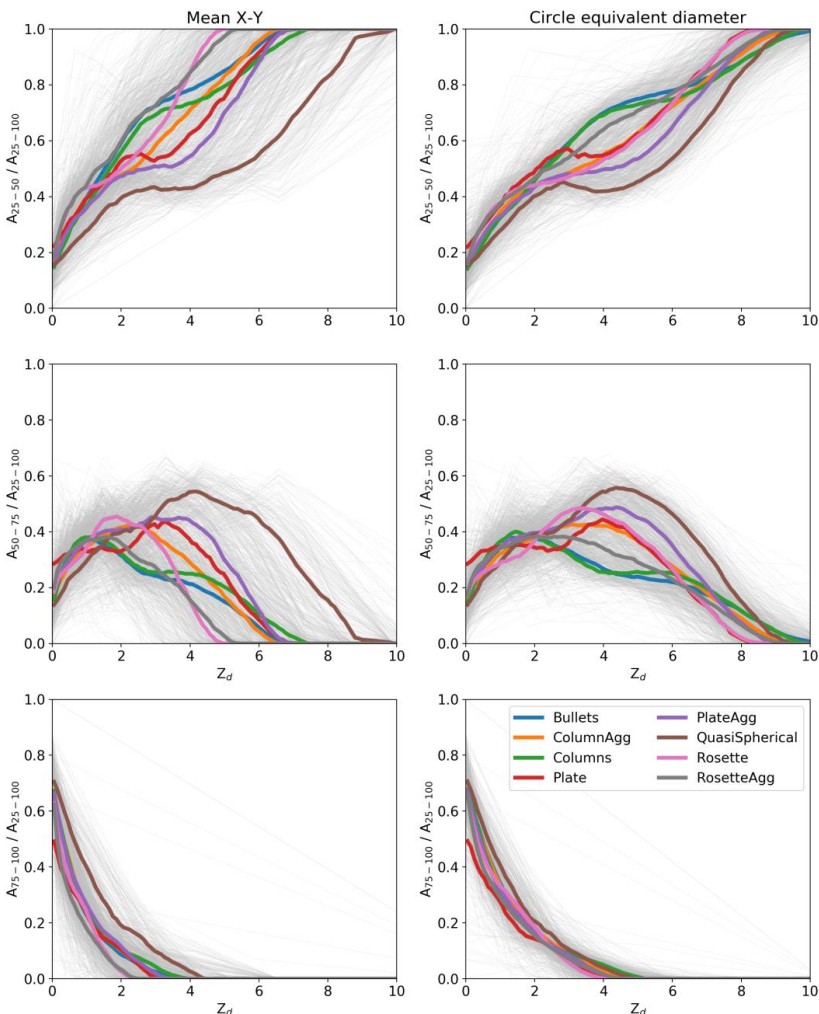

*Figure 10. Combinations of number of pixels at different greyscale ratios as a function of $Z_d$*

*for the simulated ice crystal images. Left panels show plots where mean X-Y is used as the*

*sizing metric, while the right panels use circle equivalent diameter.*





**3.5   Habit recognition**
The shape of ice crystals is a key microphysical parameter impacting cloud radiative properties
in a number of ways. A variety of automatic image recognition algorithms have been applied
to OAP datasets to classify particles into different habits (Korolev & Sussman, 2000; Crosier
et al., 2011; Praz et al., 2018). These algorithms typically rely on geometrical features extracted
from OAP images that have characteristic values for specific habits. These characteristic values
are usually determined by manually classifying images into habits. These images are then used
to set thresholds or train machine learning algorithms to automatically classify new images. For
example, Crosier et al. (2011) used the following ratio to discriminate between ice crystals and
liquid droplets:
$$Circularity = \frac{P^2}{4\pi A}$$

12                                                                                  Equation 4

where P is the particle perimeter, and A is the particle area including any internal void. Crosier
et al. (2011) used a threshold of 1.25 to discriminate between these two categories. When
images are manually selected to train habit recognition algorithms only images that can be
identified 'by-eye' as a specific habit will be included. For OAPs this is likely to be images that
are 'in-focus'. However, the shape of an OAP image and therefore the geometrical features that
are used in habit recognition algorithms depend on where in the probe's sample volume a
particle is detected. For example, Figure 3 shows a simulated 190 µm rosette at different
distances from the object plane. It is only in the top left panel (Z=0) that it can be identified as
a rosette from its image alone. Figure 11 shows how this particle's circularity changes with Z
and $Z_d$. At Z=0 its circularity is near 4, while at Z=20 mm it is near 1 and may be confused with
a spherical droplet. Figure 11 demonstrates that the measured particle shape is highly dependent
on the position in the sample volume $Z_d$ (and Z) with the circularity decreasing by a factor 2 by
$Z_d$=1; in comparison the particle size has only changed by 15%.
The variance in geometrical features for each habit will not only be due to natural variability in
the shape of ice crystals, but also due to their position in the sample volume when measured.
To date, this second effect has not been accounted for by habit recognition algorithms.
Therefore, currently the results of habit classification algorithms on OAP datasets cannot be
considered quantitative.



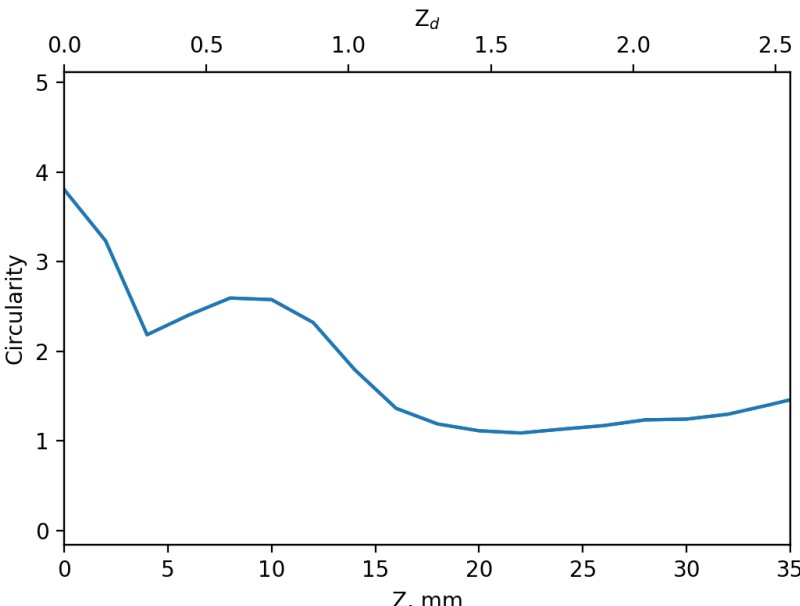

*Figure 11. The circularity (Eq. 4) of the rosette shown in Fig. 3 as function of distance from the*
*object plane Z and $Z_d$.*
**4      Methods to improve OAP size distributions**
Depending on where in the sample volume a particle is observed the OAP image size can range
between being as small as a single pixel or up to twice the true particle diameter (see Fig. 9).
Algorithms such as K07 and O19 have been derived using spherical shapes and are therefore
not directly applicable to OAP PSDs of non-spherical shapes. However, there are several
possible approaches that could be used to correct OAP ice crystal size distributions.
**4.1      Greyscale filtering**
Unlike for liquid droplets, O19 does not accurately determine $Z_d$ for non-spherical ice crystals.
We now describe a new technique to use greyscale information to remove the most severely
mis-sized ice crystals and constrain the sample volume with a reasonable uncertainty using
circle equivalent diameter as the particle sizing metric. For example, if the diffraction
simulations are filtered to only include images that have at least one pixel with a greater than a





75% drop in light intensity (Fig. 7) then the median position where particles are no longer
visible (using a 50% intensity threshold) is $Z_d = 4.6$ (interquartile range 1.1 in $Z_d$). This removes
the fragmented images that begin to occur at approximately $|Z_d| > 6$. The median ratio $D/D_0$ for
$Z_d < 4.6$ is 1.2 (interquartile range = 0.1), however, particles may still be oversized by
approximately 40% even with this filter applied (Fig. 7). Other greyscale thresholds may be
used to provide a more or less restrictive DoF constraint. Table 2 shows the median
(interquartile range) c values for various greyscale thresholds between 65 and 85%. Using a
65% threshold the median c value is 6.2 (interquartile range = 1.3), while for 85% it is 3.2
(interquartile range = 0.9). It should be noted that the lower the greyscale threshold the higher
the probability of a fragmented image being observed, and the small particle concentration
being biased.

| Greyscale intensity threshold, % | 65 | 70 | 75 | 80 | 85 |
|---|---|---|---|---|---|
| c | 6.2 (1.3) | 5.4 (1.1) | 4.6 (1.1) | 4.0 (1.2) | 3.2 (0.9) |
| $D/D_0$ | 1.2 (0.1) | 1.2 (0.1) | 1.2 (0.1) | 1.2 (0.1) | 1.2 (0.1) |

*Table 2. Median (interquartile range) depth of field c value (Eq. 1) for 1060 ice crystal images*
*using various greyscale intensity thresholds and circle equivalent diameter. The median*
*(interquartile range) ratio $D/D_0$ for $Z_d < c$ is also given.*
Figures 12 and 13 apply this new methodology to ambient measurements collected during
research flights in cirrus on 7 February 2018 and 23 April 2018. Figure 12 shows PSDs from
the CIP-15 and HALOHolo for a run at -42°C on 7 February 2018 (16:02:00 to 16:10:00 GMT).
This flight has previously been discussed by O19. Figure 13 shows equivalent PSDs for
temperatures between -47 and -40 °C collected on 23 April 2018. For both probes the particle
diameter given is the circle equivalent diameter and particles in contact with the edge of the
CIP-15 optical array have not been included in the PSD calculation. The black lines show the
CIP-15 size distribution when images are filtered to only include those with at least one pixel
at the 75% intensity threshold. This threshold significantly reduces the concentration of small
particles (<200 µm) compared to when this filtering is not applied (grey lines) and generally is

10000





in much better agreement with HALOHolo a holographic imaging probe (blue markers). This
suggests that for these cases using current data processing techniques, a significant fraction of
the ice crystal number concentration at sizes < 200 um is an artefact due to optical effects.

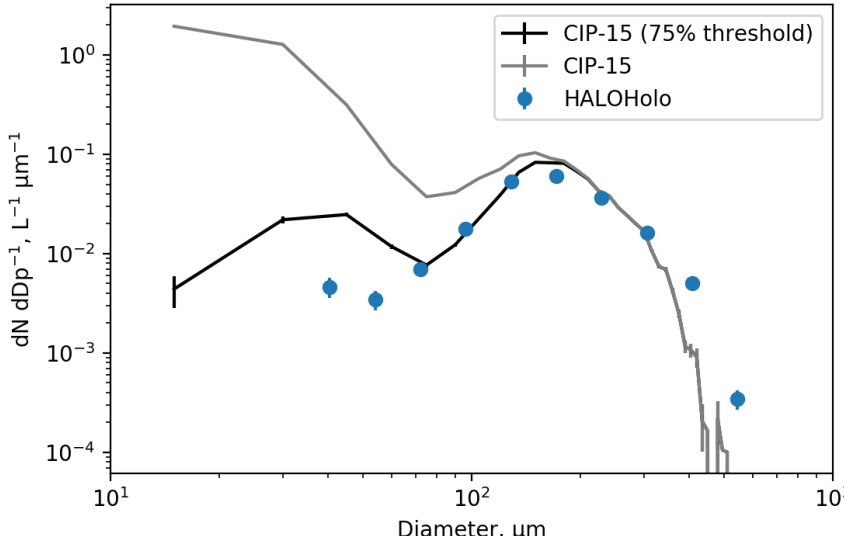

*Figure 12. Size distributions from the CIP-15 and HALOHolo for a run at -42°C on 7 February*
*2018. The black line shows the CIP-15 size distribution when images are filtered to only include*
*those with at least one pixel at the 75% intensity threshold.*

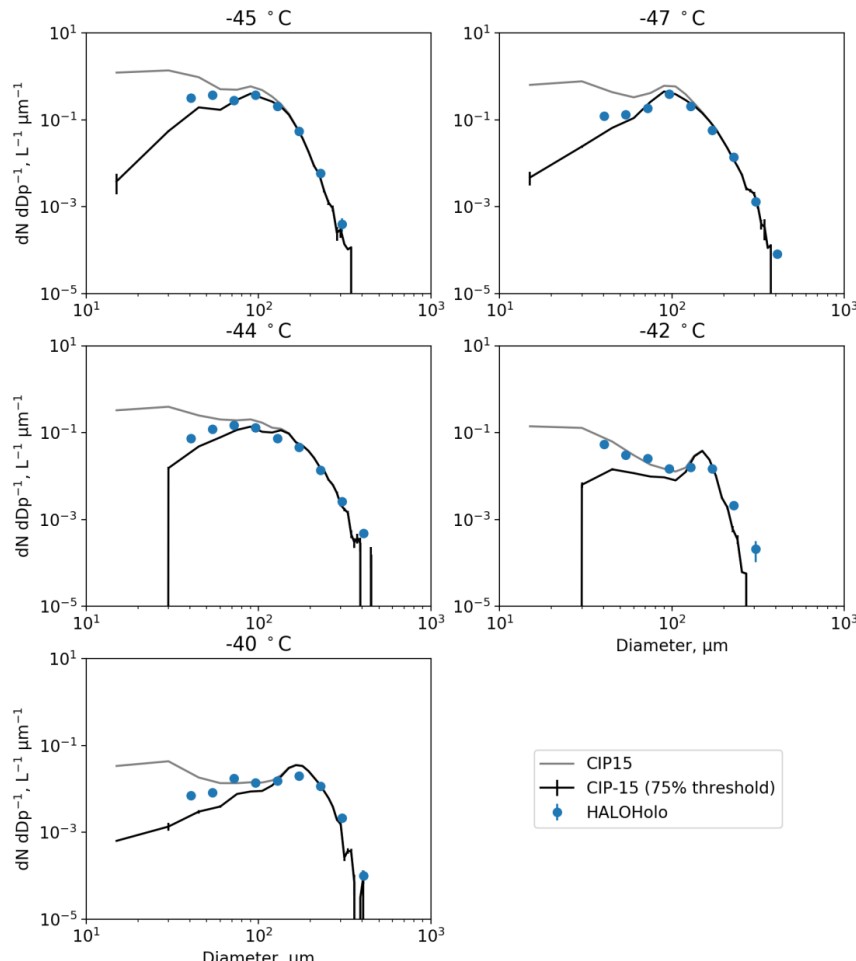

*Figure 13. Size distributions from the CIP-15 and HALOHolo for a runs between -47 and -40°C on 23 April 2018. The black line shows the CIP-15 size distribution when images are filtered to only include those with at least one pixel at the 75% intensity threshold.*



## 4.2 Stereoscopic imaging

A second method that could be used to constrain the DoF of an OAP is to use the stereoscopic imaging that is possible with the 2D-S. The 2D-S in effect consists of two OAPs (known as channels) orientated perpendicular to each other and the direction of motion of the particle/instrument. Under normal operation the probe is oriented so that one laser beam is horizontal and the other is vertical. The two lasers overlap at the centre of each channel's arms. As well as increasing sampling statistics by having two channels which can be merged/averaged, this design also allows some ice crystals to be viewed from two orientations to study their aspect ratios. In this study we use this feature to constrain the probe's DoF, which greatly limits the magnitude of diffraction artefacts, and represents the first implementation of stereoscopic analysis on an ambient OAP dataset. The 2D-S was designed so that Z= 0 on both channels is in the region where the two lasers overlap. We refer to particles observed by both channels as co-located particles. Co-located particles have tightly constrained Z position and should not be subject to significant mis-sizing due to diffraction. For the 2D-S this is likely to be true for $D_0 > 20\ \mu m$. For a hypothetical stereoscopic probe with larger optical arrays (NR) it may be necessary to restrict the distance a particle can be from the centre of the optical array.

For the case where channel 0 is used for particle sizing and channel 1 is used to constrain the particle Z position, the sample volume of co-located particles is given by,

$$SVol = TAS\left(minimum\left(\frac{cD^2}{2\lambda}, ER\right)\right)(ER - D_{CH0})$$

Equation 5

Where TAS is the true air speed, E is the number of array elements, R is the resolution of the probe, D is the measured particle diameter and $D_{CH0}$ is the particle diameter measured along the axes of the channel 0 optical array. This requires that particles in contact with the edge of the channel 0 optical array have been removed. If channel 1 is used for particle sizing instead of channel 0 then particles in contact with the edge of the channel 1 optical array are removed instead of channel 0, and $D_{CH0}$ is replaced by $D_{CH1}$ in Eq. 5.

For this method to be applicable it is important to validate that Z=0 on both channels is in the laser overlap region. If it is significantly offset this would prevent small co-located particles from being observed, since the DoF from one channel would not overlap with the optical array of the other channel. Increasingly large offsets between the channels prevent increasingly large co-located particles from being observed. It is therefore important to check that this offset is



not significant by regularly sampling in environments where small particles are present (i.e. in
liquid cloud or using a droplet generator in a laboratory as in O19).
Co-located particles could be confused with shattered particles since they are also associated
with short inter-arrival times. Figure 14 (top panel) shows a histogram of inter-arrival time for
particles on the same channel for measurements in cirrus on 7 February 2018. To minimise
shattering events, each channel was independently filtered for particles using an inter-arrival
threshold of $1\times10^{-5}$ s. It may still be possible to mistakenly detect shattered particles as co-
located particles if one shattering fragment splits into two particles, triggering each channel
simultaneously but in spatially independent parts of the sample volume. However, examination
of co-located images suggest that this is rare.
To identify co-located particles, we use the difference in arrival time between a particle on one
channel and their closest neighbour on the other channel. Figure 14 shows a histogram of co-
location times for measurements in cirrus on 7 February 2018. This distribution is bi-modal
with a larger mode centred at approximately $1\times10^{-3}$ s. and a smaller mode at $1\times10^{-7}$ s. The larger
mode is associated with the typical spatial separation between ambient particles, with its
position dependent on the particle concentration. Examining pairs of images from the smaller
mode suggests these images are the same ice crystal viewed from different orientations. Figure
15 shows example pairs of co-located images, with channel 0 images shown in yellow and
channel 1 images shown in blue. In addition to overall consistency in the geometrical shapes
between channel 0 and channel 1 images, there is also excellent consistency in the particle size
along the airspeed direction (x-axis in Figure 15) between these two channels.
Figure 14 shows that most co-located particles don't trigger both channels simultaneously
within the time resolution of the data acquisition system but are offset by a few hundred
nanoseconds. At 100 m s$^{-1}$ data slices from the detectors are acquired every $1\times10^{-7}$ s, which
corresponds to a spatial separation of 10 µm. Using the laboratory droplet generator system
described in O19, we were able to generate a continuous stream of droplets of known size,
velocity, rate, and with precise control over the position within the sample volume. These
experiments with particle velocities of 1 m s$^{-1}$ resulted in a $1\times10^{-5}$ s mode time delay in detection
events between the two channels of the 2D-S. This also corresponds to an offset of 10 um in
the sample volume in the axis of airflow through the probe (Y axis). These two sets of analysis
provide a robust independent verification of the spatial offset between the two channels of the



1    2D-S. Therefore, when considering ambient data, we classify co-located particles as those with

2    time separations less than $5 \times 10^{-7}$ s.





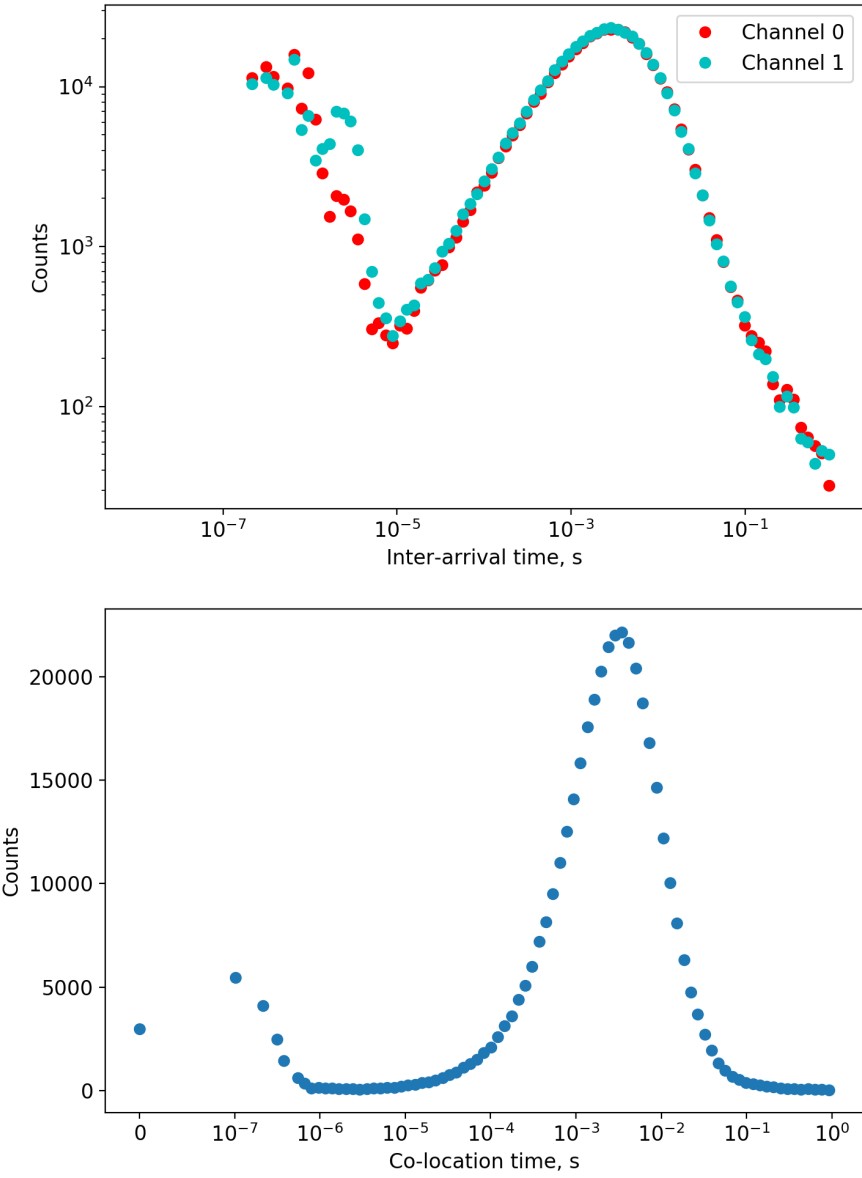



1    *Figure 14. Top panel. Histograms of inter-arrival times for particles on the same 2D-S channel*

2    *for measurements in cirrus on 7 February 2018. Bottom panel. A histogram of the difference in*

3    *arrival time between a particle on one channel and their closest neighbour on the other channel.*



10:53:06.883000 to 10:53:25.383000

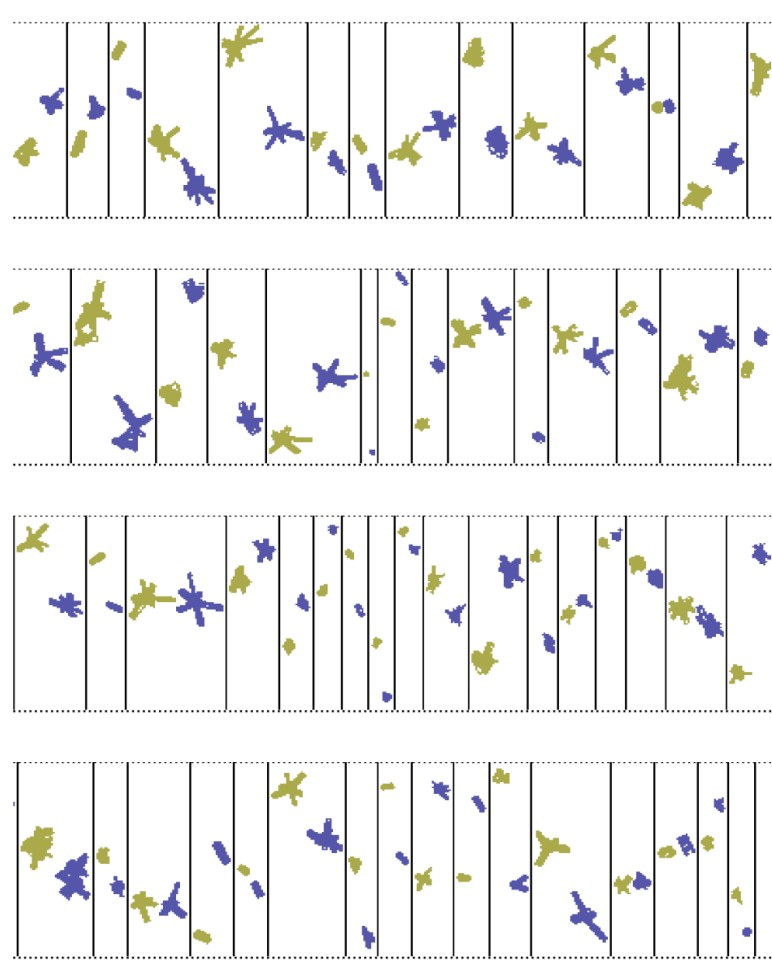

2    *Figure 15. Example ice crystals observed by both channels of the 2D-S. Images from channel*

3    *0 are shown in yellow and images from channel 1 are shown in blue.*



When determining the particle size from co-located images it is advantageous to use the largest
object in the image frame. Occasions where a single particle have been imaged as two objects
in the same image frame due to diffraction are removed by restricting the sample volume to a
narrow Z range. When sampling in environments with very high concentrations of small
particles (e.g. in liquid cloud) it is possible that two ambient particles could occur in the same
image frame. Under these circumstances using the largest particle in the image frame prevents
significant particle mis-sizing.
Figure 16 shows a comparison between PSDs collected in liquid stratus cloud at 13 °C on 17
August 2018. The grey lines show the 2D-S data for each channel using conventional data
processing protocols without constraining the DoF, while the green and red lines show PSDs
for just the co-located particles. The CDP is shown in blue. For this case no particles larger than
approximately 200 µm are present and all data processing methods are in good agreement. This
illustrates the ability of the 2D-S to detect small co-located particles.

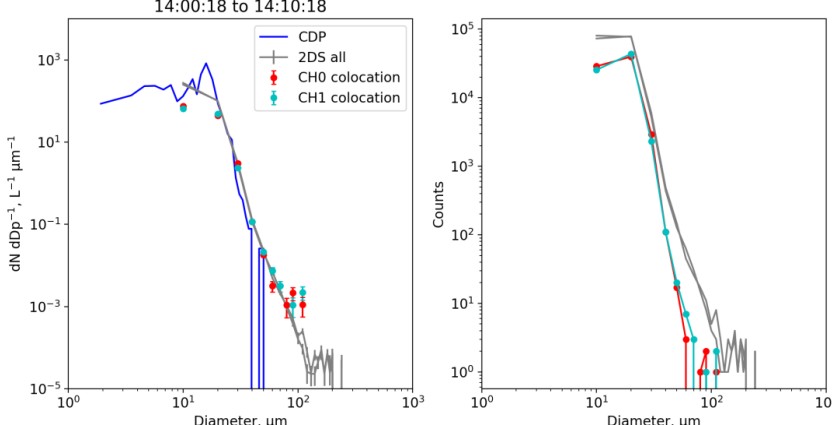

*Figure 16. Size distributions from the 2D-S and CDP for different temperatures during a*
*research flight in liquid stratus 17 August 2018 at 13 °C. The grey lines show the 2D-S data*
*using conventional data processing protocols without constraining the DoF, while the green*
*and red lines show size distributions for just the co-located particles. CDP size distributions*
*are shown in blue.*





Figure 17 shows size distributions from the 2D-S and HALOHolo for different temperatures
(averaged over ~10 minutes) during a research flight in cirrus on 11 March 2015 (see O'Shea
et al., 2016). The grey lines show the 2D-S data for each channel using conventional data
processing protocols without constraining the DoF, while the red lines show size distributions
for just the co-located particles. HALOHolo size distributions are shown in blue. For all
temperatures the conventional 2D-S data processing shows an ice crystal mode at small sizes
($< 200\,\mu m$). At warmer temperatures ($> -39\,°C$) there is also a clear second mode at larger sizes.
However, these high concentrations of small ice particles are not present in the co-located and
the HALOHolo size distributions. This suggests using only co-located particles on the dual
channel 2D-S probe is effective at removing significant biases at small particle sizes. At larger
sizes ($>300\,\mu m$) the 2D-S data processing using conventional and stereoscopic methods are in
good agreement, however the latter method is limited by sampling statistics.
Stereoscopic data processing has the advantage of removing out-of-focus artefacts that bias the
PSD at small sizes, while at larger sizes traditional processing methods offers significantly
improved sampling statistics. Therefore, a hybrid approach using stereoscopic processing for
small sizes and traditional processing methods for larger sizes is advantageous. The choice of
size threshold to switch between the two methods is dependent on the arm width of the probe
and the level of mis-sizing that is deemed acceptable. To give an idea of a suitable threshold,
we will choose a size limit that prevents all particles with $Z_d > 2$ from being included in the
PSD. The maximum Z that the 2D-S can observe a particle is Z=31.5 mm (2D-S armwidth/2)
This corresponds to a 222 µm particle at $Z_d = 2$. However, since particles can be mis-sized by
a factor 1.4 then a size threshold of 300 µm is needed to ensure that no particle with $Z_d > 2$ is
included. Figure 17 dashed lines shows 2D-S PSDs processed using this hybrid approach.

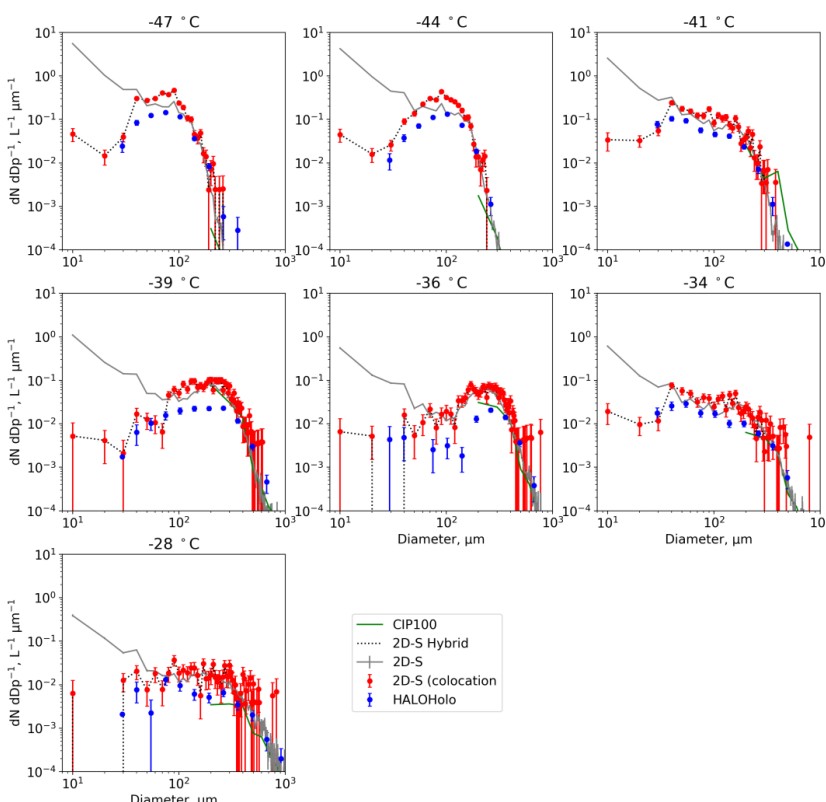

*Figure 17. Size distributions from the 2D-S and HALOHolo for different temperatures during*

*a research flight in cirrus on 11 March 2015. The grey lines show the 2D-S data using*

*conventional data processing protocols without constraining the DoF, while the green and red*

*lines show size distributions for just the co-located particles. The dashed black line shows a*

*2D-S processed using a hybrid of conventional and co-location data processing (see text for*

*details). HALOHolo size distributions are shown in blue.*



### 4.3 Other potential methods

There are several other potential methods that could be used to improve OAP PSD measurements. First reducing a probe's arm width to physically limit a distance a particle can be from the object plane would reduce out-of-focus particles. The amount the arm width would need to be decreased depends on the level of mis-sizing that is deemed acceptable for a given particle size, with more accurate sizing and smaller particles requiring smaller arm widths. However, as well as decreasing the sample volume, reducing the probe's arm width is likely to increase the proportion of shattered artefacts particles compared to ambient particles that the probe measures, since shattered artefacts are thought to cluster near the probe's arms.

Second, statistical retrievals have been applied to particle size distribution measurements where the instrument response is a distorted version of the true ambient distribution. These methods are reliant on knowing or empirically approximating the instrument function that distorts the ambient distribution. These methods have been applied to OAP measurements of spherical droplets (Korolev et al., 1998; Jensen & Granek, 2002). For non-spherical particles the distortion function is dependent on the ice crystal habits present and therefore the derived size distributions would have greater uncertainty, unless the particle shape is known a priori. However, this methodology may still result in an acceptable level of uncertainty if circle equivalent diameter is used, since its intra- and inter-habit $D/D_0(Z_d)$ variance is smaller than for the mean X-Y and maximum diameter.

## 5 Implications for small ice crystal observations

In-situ measurements of ice clouds have consistently observed a mode in particle size distributions at small sizes ($< 200\,\mu m$). This would imply that ice nucleation occurs at all cloud levels, since small ice particles would rapidly grow in regions of ice super-saturation or sublime in sub-saturated regions. Particle shattering on the leading edge of a probe has previously been identified as a possible explanation (Korolev et al., 2005; Korolev et al., 2011). However, the impacts of shattering are thought to have been minimised by modifying the leading edges of probes (Korolev et al., 2013) and using particle inter-arrival time algorithms (Field et al., 2006). Yet even with these improved measurements a small ice mode has been found to be ubiquitous in ice cloud observations (McFarquhar et al., 2007; Jensen et al., 2009; Cotton et al., 2013; Jackson et al., 2015; O'Shea et al. 2016).



This work has shown that depending on where in the OAP sample volume a particle is observed
its image size can be as small as a single pixel or up to a 200% overestimate of the true particle
diameter (see Fig. 9). Only a relatively small proportion of undersized larger particles are
required to generate a significant bias in number concentration at small sizes ($< 200\,\mu m$) due
to the size dependence of the DoF (Eq. 1) (O19). We have tested two methods that could be
used to remove out-of-focus artefacts: greyscale filtering (Sect. 4.1) and stereoscopic imaging
(Sect. 4.2). Both methods either remove or significantly reduce the concentration of small ice
crystals observed in specific cirrus cloud cases (Figures 12, 13 and 17).
To further explore the impact OAP mis-sizing has on the measured PSD shape we use the results
from the AST model. Consider the ambient ice crystal PSD $N(D_0)$ with units $L^{-1}\,\mu m^{-1}$. If this
distribution is observed by an OAP with size dependent sample volume $SVol(D_0)$ (units: $L^{-1}\,s^{-1}$
$^{1}$, Eq. 2) then the number of ice crystals observed by the probe as a function of true particle
diameter $C(D_0)$ (units: $\mu m^{-1}\,s^{-1}$) is given by,

$$C(D_0) = N(D_0)SVol(D_0)$$

15                                                                          Equation 6

The number of ice crystals observed as a function of the measured diameter $C(D)$ is given by,

$$C(D) = M(D, D_0) \cdot C(D_0)$$

18                                                                          Equation 7

$M(D, D_0)$ is an E x E matrix, where E is the number of detector elements. Each row of $M(D,$
$D_0)$ is the probability distribution that a particle of measured size D has true size $D_0$. These
probabilities are dependent on the particle shape, the particle sizing metric, probe characteristics
(e.g. armwidth, laser wavelength) and the data processing protocols used (e.g. greyscale
filtering, co-location). The PSD observed by the probe $N(D)$ can then calculated by,

$$N(D) = \frac{C(D)}{SVol(D)}$$

25                                                                          Equation 8

The probe armwidth limits the maximum $Z_d$ that a particle of given $D_0$ can be observed. By
choosing an armwidth it is possible to calculate a probability distribution function of possible
D for each $D_0$ from one of the $D/D_0(Z_d)$ relationships shown in Fig. 9. For our example, we use
an armwidth of 70 mm and the median $D/D_0(Z_d)$ relationship for rosettes. We calculate $M(D,$





D$_0$) for two cases: when mean X-Y and circle equivalent diameter are used as the particle sizing
metric. To represent the true ambient distribution, we use three different gamma distributions
that all have the form,
$$N(D) = N_0 D^\mu e^{-\lambda D}$$

5                                                                          Equation 9

Figure 18 shows three combinations of the coefficients $\mu$, $\lambda$ (cm$^{-1}$) and N$_0$ (L$^{-1}$ cm$^{-1}$). Left panel
shows plots using mean X-Y diameter and the right panels shows circle equivalent diameter.
The ambient PSDs (blue lines) are compared to simulated OAP observations using different
data processing methodologies. The grey lines represent an OAP with armwidth of 70 mm using
conventional data processing methods. The red markers represent a 2D-S using only co-located
particles, which has the effect of limiting the maximum Z a particle can be observed to 0.64
mm. The blue markers show simulated OAP measurements from a greyscale probe with 70 mm
arm width when the data has been filtered to only include particles that have at least one pixel
with a greater than 75% decrease in light intensity.
It should be noted that these simulated distributions only include mis-sizing due to diffraction
and do not include other sources of OAP measurement uncertainty (e.g. counting statistics).
Counting statistics will be responsible for a larger uncertainty for the co-located PSDs
compared to conventional data processing methods.
Figure 18 top panels show an ambient distribution (blue lines) dominated by small particles ($\mu$=
-1, $\lambda$ =1000 cm$^{-1}$ and N$_0$ = 10 L$^{-1}$ cm$^{-1}$), with concentrations increasing with decreasing size
over the displayed size range 10 to 1280 $\mu$m, which is representative of modern OAPs. The
grey lines show the simulated OAP observations of this PSD, which have a similar
characteristic shape. The total particle concentration observed by the simulated OAP over the
size range 10 to 1280 $\mu$m is 3% and 13% higher than the true PSD using mean X-Y and circle
equivalent diameter, respectively. Figure 18 top left panel show the PSD that a 2D-S would
observe when only co-located particles are included (red markers). The total particle
concentration from the co-located PSD differs from the ambient distribution by less the one
percent. The total particle concentration when greyscale filtering is applied is 2% lower that the
true distribution.
Figure 18 middle panels show an ambient distribution with mode near 100 $\mu$m particles ($\mu$ = 2,
$\lambda$ =200 cm$^{-1}$ and N$_0$ = 1x10$^4$ L$^{-1}$ cm$^{-1}$). The simulated OAP PSDs have significantly different





shape with much higher concentrations of particles <100 µm. Here the OAP overestimates the
total particle concentration over the size range 10 to 1280 µm by 74% and 80% using mean X-
Y and circle equivalent diameter, respectively. When stereoscopic imaging is used to constrain
the OAP sample volume (red lines) the small particle mode is removed. The true and simulated
OAP total particle concentration differ by < 1%. Greyscale filtering again removes the small
particle mode, though underestimates the total particle concentration by 11%.
Figure 18 bottom panels show an ambient PSD with mode near 400 µm particles ($\mu = 4$, $\lambda = 100$
$cm^{-1}$ and $N_0 = 1x10^6$ $L^{-1}$ $cm^{-1}$), like the previous case the simulated OAP PSD significantly
overestimates the small particle concentration. The simulated OAP PSD is bi-modal, while the
true PSD is mono-modal. However, in this case the artificial small particles contribute a
relatively small proportion to the total number concentration in the 10 to 1280 µm size range,
as a result the simulated OAP only overestimates this by 4% using both particle size metrics.



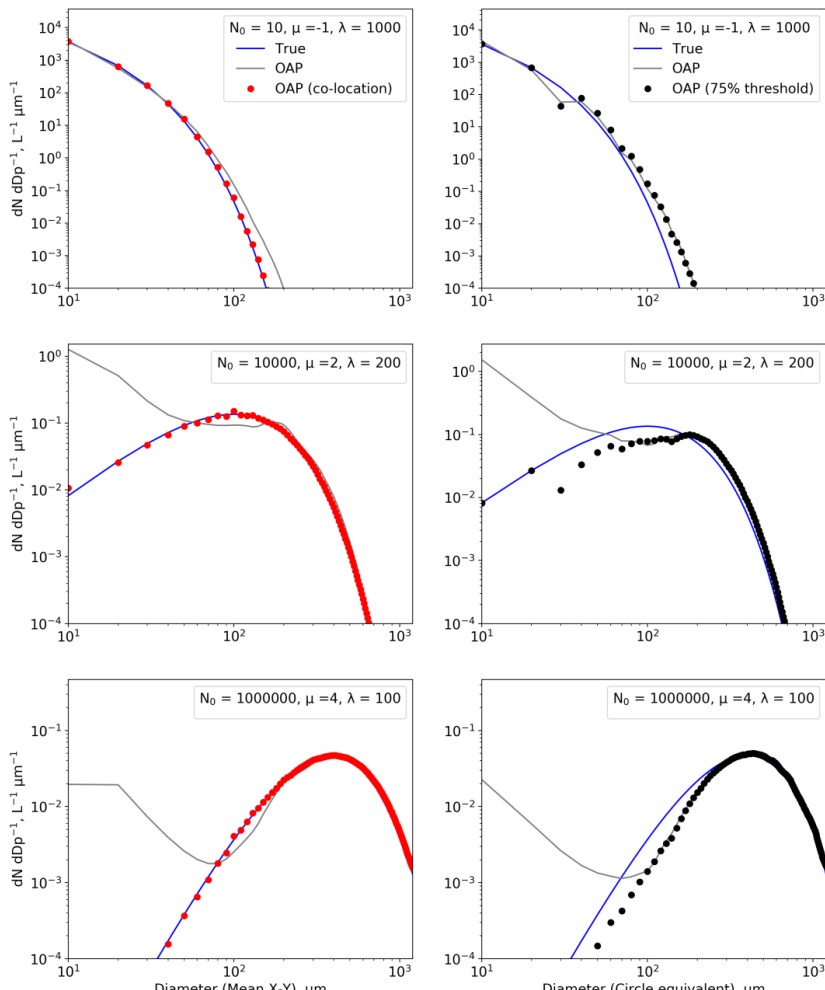

*Figure 18. Simulations of OAP measurements of different gamma PSDs (blue lines). The*

*coefficients μ, λ (cm⁻¹) and N₀ (L⁻¹ cm⁻¹) for each gamma PSD are shown in text boxes. Left*

*panel shows plots using mean X-Y diameter and the right panels shows circle equivalent*

*diameter. The grey lines show simulated OAP PSDs with armwidth of 70 mm if all the particles*





*are rosettes. The red markers show simulated 2D-S measurements using only co-located*
*particles, which has the effect of limiting the maximum Z a particle can be observed to 0.64*
*mm. The blue markers show simulated OAP measurements from a greyscale probe with 70 mm*
*arm width when the data has been filtered to only include particles that have at least one pixel*
*with a greater than 75% decrease in light intensity.*
A significant amount of our understanding of clouds microphysics is based on OAP
measurements, with the small particle artefact being present and manifesting in some manner.
This includes how PSDs are parameterised in numerical models and remote sensing retrievals.
Generally in the literature some formulation of exponential or gamma funtion has been used to
represent ice crystal PSDs for observation or modelling studies (e.g. Cazenave et al., 2019;
Delanoë et al., 2005; 2014; Field et al., 2007; Heymsfield et al., 2013; McFarquhar, &
Heymsfield, 1997). These functions and the coefficients that are used in the literature all result
in the highest ice cyrstal concentrations at the smallest sizes. For example, Field et al. (2007)
describes a parameterisation based on OAP measurements that is widely used by the passive
and radar remote sensing communities (e.g. Mitchell et al., 2018; Sourdeval et al., 2018;
Ekelund et al., 2020; Eriksson et al., 2020; Fontaine et al., 2020). It describes a characteristic
ice crystal PSD that can be used to calculate moments of a PSD when the ice water content is
known. The functional form of the parameterisation consists of the summation of a gamma and
expontential distribution.
Figure 19 shows a comparion between the 2D-S PSD for 11 March 2015 and the Field et al.
(2007) parameterisations for tropical (Eq. 10) and mid-lattiude (Eq. 11) ice clouds.
$$N(D)\frac{M_3{}^3}{M_4{}^2} = 152\, e^{-12.4x} + 3.28\, x^{-0.78}e^{-1.94x}$$
Equation 10
$$N(D)\frac{M_3{}^3}{M_4{}^2} = 141\, e^{-16.8x} + 102x^{2.07}e^{-4.82x}$$
Equation 11



where the number concentration (N(D)) and diameter are normalised using the second (M2)
and third (M3) moments of the PSD and x is equal to $DM_2/M_3$. The 2D-S PSD in Fig. 19 has
been calculated using only co-located particles for D <300 µm and all particles for D > 300 µm.
Both the tropical and mid-latitude parameterisations show rapidly increasing concentrations
with decreasing size. At larger sizes the 2D-S and these parameterisations are in good
agreement, while they diverge at smaller sizes. The green line in Fig. 19 shows the gamma
component of the mid-latitude F07 parameterisation (Eq. 12), which is in much better
agreement with the observations at small sizes.
$$N(D)\frac{M_3{}^3}{M_4{}^2} = 102x^{2.07}e^{-4.82x}$$

11                                                              Equation 12

This work suggests that the data used for derived PSDs parameterisations is subject to
significant artefacts. As a result, the parameterisations are likely to have incorrect fundamental
shape for ice cloud PSDs. The impacts of these artefacts can be expected to propagate to
inaccuracies in remote sensing retrievals, which will be assimilated into weather forecast
models, and to incorrect radiative properties due to a bias towards small particle sizes. Future
work is needed to quantify the impact on retrievals and our understanding of ice microphysics
and cloud radiative properties using the improved measurement methodologies presented in
this paper.

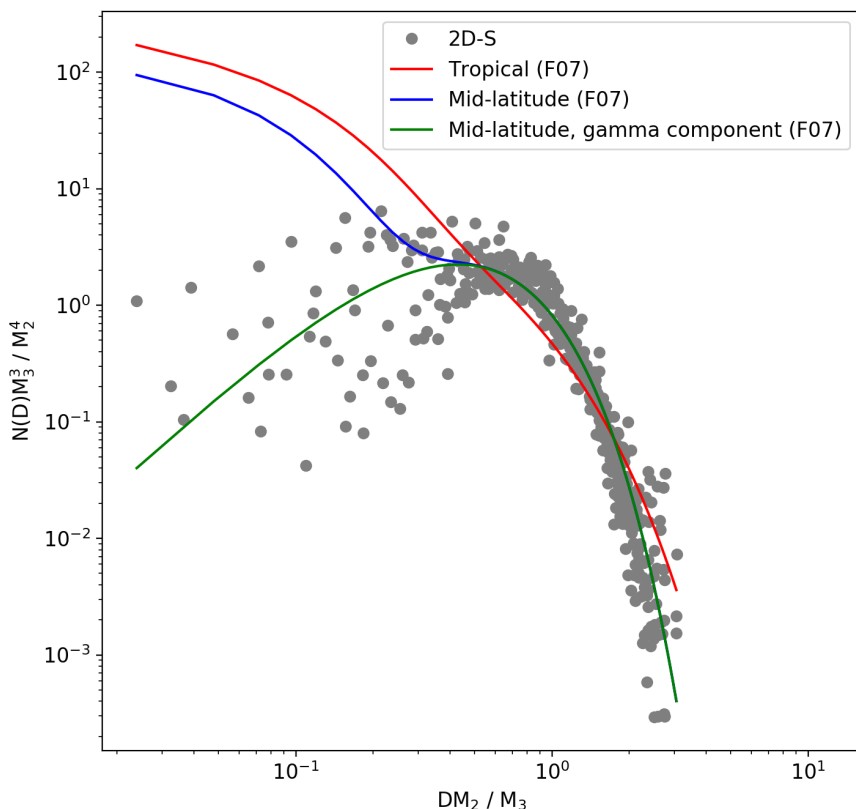

*Figure 19. Comparison between 2D-S size distributions of co-located particles from a research*
*flight in cirrus on 11 March 2015 and Field et al. (2007) parameterisations for tropical and*
*mid-latitude ice clouds.*
**6    Conclusions**
This paper quantifies the optical response of OAPs to non-spherical particles for understanding
ice crystal observations, expanding the work of O19. We make the following comments and
recommendations on the use of OAP data:



•   The shape and size of an OAP image depends significantly on where in the OAP sample
volume a particle is observed. Particles < 200 µm are the most significantly mis-sized.
The measured size of a particle can range between being as small as a single pixel up to
being as large as a 200% overestimate of the true particle.

•   Particle mis-sizing and the size dependence of the OAP sample volume causes an
artefact which results in systematic overestimate of small ice (< 200 µm) concentrations.
The persistent mode of small sizes observed in many previously studied cases is likely
artificial. However, the importance of this artefact is strongly influenced by the true
shape of the ambient PSD.

•   Algorithms to correct OAP size distributions such as K07 and O19 that have derived
using spherical particles are not applicable to non-spherical ice crystal images without
significant uncertainty.

•   New methods that may be used to filter OAP ice crystal size distributions were tested,
including filtering using grayscale, and the use of stereoscopic imaging.

•   For greyscale instruments (such as the CIP-15), filtering images so that they must
include one pixel with at least a 75% decrease in detector intensity removes the most
severely fragmented particles near the edge of the DoF. This approach constrains the
DoF to c = 4.6 (interquartile range 1.1) using circle equivalent diameter.

•   Using the stereoscopic imaging that is possible with the 2D-S can constrain the sample
volume to only 'in-focus' images. A hybrid approach using stereoscopic processing for
small sizes and traditional processing methods for larger sizes is advantageous, as it
limits any negative impacts on sample volume and therefore counting statistics. The
choice of size threshold to switch between the two methods is dependent on the arm
width of the probe and the level of mis-sizing that is deemed acceptable. For the 2D-S
we suggest that 300 µm is a suitable threshold for particle sizing using the mean X-Y
diameter.





•   These new methodologies were tested using data from three research flights sampling
cirrus. In these cases, they significantly improved agreement with a holographic
imaging probe compared to conventional data processing protocols and either removed
or significantly reduced the concentration mode at small particle sizes (<200 µm). This
raises the question over the interpretation of many existing datasets such as those used
to parameterise PSDs (e.g. Delanoë et al., 2005; 2014; Field et al, 2007), and the
persistent observation of small particles throughout the entire vertical extent of ice
clouds which has been difficult to reconcile with concepts of ice nucleation.

•   Past datasets from OAPs need to be revisited, where possible the filtering and sample
volume adjustments described in this paper should be applied. The impact these
corrections have on how PSDs are parameterised in numerical models; remote sensing
retrievals and radiative calculations of ice clouds need to be examined.

**Data availability**
The data presented here can be provided on request to the contact author.

**Acknowledgements**
We would like to thank Thibault Vaillant de Guélis for his help with the AST model. We are
grateful to Jacob Fugal for assistance with HALOHolo. The authors wish to thank Hannakaisa
Lindqvist (CSU) for making available her CPI training data set. Airborne data were obtained
using the BAe-146-301 Atmospheric Research Aircraft (ARA) flown by Directflight Ltd and
managed by the Facility for Airborne Atmospheric Measurements (FAAM), which is a joint
entity of the Natural Environment Research Council (NERC) and the Met Office. The CIP-15s
were provided by the National Centre for Atmospheric Science and FAAM. The National
Centre for Atmospheric Science provided support for the ice crystal analog experiments. This
work was supported by the NERC grants NE/P012426/1 and NE/L013584/1.



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
