# Peer review of "Characterising optical array particle imaging probes: implications"

_Atmospheric Measurement Techniques, 2020_

## Referee Comment (RC1) · Anonymous Referee #2 · 22 Sep 2020

As pointed out in the manuscript, in situ measurements of ice particle size distributions (PSDs) have been made to a large extent by optical array probes (OAPs) and they are still in frequent use. These probes have indicated high n$\mu$mber concentrations of particles at sizes below 100 $\mu$m. It has been shown that this is partly an artefact of shattering of larger particles. Measures to remove the impact of shattering have been developed. The question is if PSDs derived from these optical probes now are roughly correct, or if there still exist basic issues?

It seems that this study is motivated by remaining discrepancies between PSDs derived by OAPs and another type of sensor, HALOHolo, observed for some flights by the

[Figure]

FAAM Bae-146 research. By developing and applying measures to remove the impact of particles out of focus, that then are mis-sized, the discrepancies are largely removed.

The analysis behind the measures to avoid mis-sizing is ambitious and appears to have been performed in a solid manner. The change in PSDs for particles below 30 $\mu$m varies but can reach orders of magnitude. Such changes in PSDs have broad implications for the radiative transfer involving ice particles, both inside atmospheric models and inversions of observations.

The overall analysis has intrinsic value and is sufficient to make the article highly relevant. However, it would of course be nice to know if we now can trust PSDs derived from OAPs (if measures to avoid mis-sizing are applied)? The authors do not claim this, but the much-improved agreement with HALOHolo still hints in this direction. Anyhow, the agreement with HALOHolo works as a proof of concept and that makes the quality of the HALOHolo PSDs important.

In s$\mu$mmary, I find the manuscript very interesting and having an overall high quality.

General comments:

The manuscript is relatively long. There is already a supplement and I suggest moving some figures to the supplement. For example, despite it is nice to see the hardware, Figure 2 did not help me to understand the measurement setup. It is sufficiently well described in the text. It could suffice to just show one of Fig, 5, 6, and 7 in the manuscript. Maybe the same for Fig 12 and 13.

On the other hand, I would like to see an extension around HALOHolo. As discussed above, the accuracy of HALOHolo PSDs is quite crucial for the final interpretation of the results. Could HALOHolo in any way miss particles of 10-30 $\mu$m size? Maybe a naive question, but I don't know anything about HALOHolo and the manuscript does not give any hints. Adding some reference(s) is a basic demand, but I would prefer to also see a critical discussion.
I can not find any clear rule on how to name variables in the AMT guideline, but I would still suggest using letters for variables instead of acronyms. A basic problem with using acronyms is found in e.g. Eq 2. Is ER here a variable, or the product between E and R? Maybe OK in an equation but can cause severe confusion on the text (e.g. page 27, line 15). In any case, variables shall be typeset in a consistent manner, and not be different between text and equations.

Section 3 should be renamed to "Results and discussion".

Smaller issues:

The text is in general very clear. The only part I could not follow is the first paragraph on page 33. The problem is likely the start sentence.

More or less the same for figures. I don't understand the right panel of Figure 16 (also on page 33!). Why is it needed? And are the two panels really consistent? Further, CDP seems to disagree above 50 $\mu$m, in contrast to what is written at line 12.

Some even smaller issues I leave for the copy editing.

---

## Referee Comment (RC2) · Anonymous Referee #1 · 23 Sep 2020

Review of "Characterising optical array particle imaging probes: Implications for small ice crystal concentrations" by O'Shea et al.

Recommendation: Requires major revision after which its suitability for publication can be reassessed

The subject matter of this manuscript is timely and within the scope of AMT. It has been long recognized that there are significant uncertainties in the concentrations of small ice crystals that are measured by optical array probes due not only to the possibility of ice crystal shattering, but also because there is a small, poorly defined and dimensional-dependent depth of field. This paper attempts to improve upon the determination of a probe's sample volume by using grayscale image analysis and co-location using stereoscopic imaging. Although the former has already been treated in the literature, the later is especially a novel contribution that deserves publication. Further, the manuscript is well written and the figures are of high quality. As such, it appears that the paper should eventually appear in AMT. However, as the conclusions of this study can be far-reaching, it is important that the technical details of the study have the highest quality. There are some overarching concerns about some of the analysis and some misinterpretations of previous studies that should be corrected before this paper is accepted.

First, the Korolev (2007) technique was never designed to work on non-spherical particles so the application of this technique to non-spherical particles is not appropriate here, even if prior studies have applied the technique to non-spherical particles.

Second, for the determination of the probe sample volume, there are optical reasons why the diameter in the direction of the photodiode array should be used in the consideration of the depth of field of the instrument. If this definition is not used, a proper depth of field dependence on particle habit/placement in the array cannot be derived.

Third, the paper exaggerates the implications of the study for the parameterizations or representations of small ice crystals. Most previous studies have specifically noted that there are large uncertainties in quantifying the contributions of particles with dimensions smaller than 150 micrometers due to small and poorly defined depths of field for small particles (going back to a study of Baumgardner and Korolev 1997 that has been cited many times). This study is not even referenced here! This study seems to be using a threshold size of 200 um rather than 150 um, so there is a bit of a difference here. But, the findings of the manuscript should be better placed in the context of other studies that have already been conducted as otherwise the implications of this study are overexaggerated.

I'm not sure that the data availability statement meets the threshold required by the

journal. In general, the data should be available at a publicly accessible web site rather than only on request to the contact author. If this study is going to have far-reaching implications, these data should be more openly available for others to test their algorithms with. I'll leave it for the Chief Editor to decide if this statement is adequate.

―――――――――――――――

---

## Author Comment (AC1) · 1 Dec 2020

We would like to thank both reviewers for their constructive comments about our manuscript. We now address their comments individually. For clarity referee comments are shown in red and our responses in black.

For context, the comments from referee 2 were received in pre-review and have been at least partially addressed before publication in AMTD. The referee may not have seen the changes in the version published in AMTD.

Referee 1

As pointed out in the manuscript, in situ measurements of ice particle size distributions (PSDs) have been made to a large extent by optical array probes (OAPs) and they are still in frequent use. These probes have indicated high number concentrations of particles at sizes below 100 μm. It has been shown that this is partly an artefact of shattering of larger particles. Measures to remove the impact of shattering have been developed. The question is if PSDs derived from these optical probes now are roughly correct, or if there still exist basic issues?

It seems that this study is motivated by remaining discrepancies between PSDs derived by OAPs and another type of sensor, HALOHolo, observed for some flights by the FAAM Bae-146 research. By developing and applying measures to remove the impact of particles out of focus, that then are mis-sized, the discrepancies are largely removed.

The analysis behind the measures to avoid mis-sizing is ambitious and appears to have been performed in a solid manner. The change in PSDs for particles below 30 μm varies but can reach orders of magnitude. Such changes in PSDs have broad implications for the radiative transfer involving ice particles, both inside atmospheric models and inversions of observations.

The overall analysis has intrinsic value and is sufficient to make the article highly relevant. However, it would of course be nice to know if we now can trust PSDs derived from OAPs (if measures to avoid mis-sizing are applied)? The authors do not claim this, but the much-improved agreement with HALOHolo still hints in this direction. Anyhow, the agreement with HALOHolo works as a proof of concept and that makes the quality of the HALOHolo PSDs important. In summary, I find the manuscript very interesting and having an overall high quality.

General comments:

The manuscript is relatively long. There is already a supplement and I suggest moving some figures to the supplement. For example, despite it is nice to see the hardware, Figure 2 did not help me to understand the measurement setup. It is sufficiently well described in the text. It could suffice to just show one of Fig, 5, 6, and 7 in the manuscript. Maybe the same for Fig 12 and 13.

We are aware that the paper is relatively long, however we would prefer to include these plots in the main paper. The level of agreement between the laboratory experiments and model is very important when considering the strength of the conclusions that are drawn from the model simulations later in the paper. Including these plots in the main paper makes them more easily accessible to the reader and we expect more widely referred to.

On the other hand, I would like to see an extension around HALOHolo. As discussed above, the accuracy of HALOHolo PSDs is quite crucial for the final interpretation of the results. Could HALOHolo in any way miss particles of 10-30 μm size? Maybe a naive question, but I don't know anything about HALOHolo and the manuscript does not give any hints. Adding some reference(s) is a basic demand, but I would prefer to also see a critical discussion.

The following sentences have been added to Sect 4.1 discussing HALOHolo:

"HALOHolo's sample volume is not as strongly dependent on particle size as it is for OAPs. However as described earlier, measurements of small particles from HALOHolo are limited by noise in the background image. For a complete description of the HALOHolo data processing and quality control procedures see Schlenczek (2017). HALOHolo uses supervised machine learning to discriminate real particles from artefacts due to noise in the background image. However, it is possible that small particles could be misclassified as artefacts or vice versa, and as a result HALOHolo could either underestimate or overestimate the small ice concentration. For particles > 35 μm it is estimated that the probe's detection rate is >90% and previous work has shown excellent agreement with a CDP in liquid clouds (Schlenczek, 2017). However, HALOHolo PSDs should not be considered the true PSD, but rather another piece of evidence that suggests for these cases OAPs overestimate small ice concentrations using current data processing techniques."

I can not find any clear rule on how to name variables in the AMT guideline, but I would still suggest using letters for variables instead of acronyms. A basic problem with using acronyms is found in e.g. Eq 2. Is ER here a variable, or the product between E and R? Maybe OK in an equation but can cause severe confusion on the text (e.g. page 27, line 15). In any case, variables shall be typeset in a consistent manner, and not be different between text and equations.

The abbreviations DoF (depth of field), SVol (sample volume), and TAS (true airspeed) are widely used in the literature when presenting the OAP sample volume equations (for example the review paper McFarguhar et al., 2017). To maintain consistency with previous work we would prefer to continue using these abbreviations. ER is the product of E and R, both E and R are defined immediately after the equation.

Note there was a mistake in the effective array width in equation 2 and 5. The effective array width should be given by R(E-1)-D, this is corrected in the revised manuscript.

Section 3 should be renamed to "Results and discussion".

Done

Smaller issues: The text is in general very clear. The only part I could not follow is the first paragraph on page 33. The problem is likely the start sentence.

This sentence concerns a side point and has been removed in the revised manuscript.

More or less the same for figures. I don't understand the right panel of Figure 16 (also on page 33!). Why is it needed? And are the two panels really consistent?

The left panel shows the particle size distribution and the right panel shows instrument counts versus size for the same period. The right panel is not needed and has been removed in the revised manuscript. Note that Fig 16 and 17 have updated after removing a small bug in the stereo sample volume calculation, there are no change in the results/conclusions from these plots.

Further, CDP seems to disagree above 50 μm, in contrast to what is written at line 12.

This has been corrected in the revised manuscript to read:

"All data processing methods are in good agreement up to 100 μm. For larger sizes the measurements using the co-located particles are limited by counting statistics due to the low concentration of these particles."

Some even smaller issues I leave for the copy editing.

Referee 2

Recommendation: Requires major revision after which its suitability for publication can be reassessed

The subject matter of this manuscript is timely and within the scope of AMT. It has been long recognized that there are significant uncertainties in the concentrations of small ice crystals that are measured by optical array probes due not only to the possibility of ice crystal shattering, but also because there is a small, poorly defined and dimensional-dependent depth of field. This paper attempts to improve upon the determination of a probe's sample volume by using grayscale image analysis and co-location using stereoscopic imaging. Although the former has already been treated in the literature, the later is especially a novel contribution that deserves publication. Further, the manuscript is well written and the figures are of high quality. As such, it appears that the paper should eventually appear in AMT. However, as the conclusions of this study can be far-reaching, it is important that the technical details of the study have the highest quality. There are some overarching concerns about some of the analysis and some misinterpretations of previous studies that should be corrected before this paper is accepted.

First, the Korolev (2007) technique was never designed to work on non-spherical particles so the application of this technique to non-spherical particles is not appropriate here, even if prior studies have applied the technique to non-spherical particles.

In the manuscript we state that Korolev (2007) was designed for spherical particles. However, in the absence of an alternative it has also been applied to non-spherical particles (e.g. Davis et al., 2010). This work tests and discusses the efficacy of this approach. The following text is from Sect 3.2 of the manuscript:

"The K07 approach was derived by considering Fresnel diffraction from opaque discs and has only been tested for images of spherical droplets. However, in the absence of an alternative previous studies have applied K07 to images of ice crystals (e.g. Davis et al., 2010)."

We conclude in Sect. 3.2 that:

"For a number of habits (rosette, plate, quasi-spherical, rosette-aggregate and plate-aggregate) K07 reduces the number of over-sized particles across most of the DoF. For bullets, columns and column aggregates K07 has minimal impact on the probe sizing. For all habits, K07 is not able to remove the small image fragments that occur when a particle is near the edge of the DoF."

Second, for the determination of the probe sample volume, there are optical reasons why the diameter in the direction of the photodiode array should be used in the consideration of the depth of field of the instrument. If this definition is not used, a proper depth of field dependence on particle habit/placement in the array cannot be derived.

We are only aware of a few publications that have determined the OAP depth of field for non-spherical particles (e.g. Vaillant de Guélis et al., 2019; Gurganus, & Lawson, 2018), these studies only focused on a limited number of particle shapes and specific sizes. They do not present general rules for a range of particle shapes and sizes. This information is needed to calculate accurate particle concentrations from field measurements in ice clouds. Wu et al. (2016) show that ice cloud PSDs have high sensitivity to the choice of ice crystal size metric. However, we are not aware of any previous study that examines the suitability of different definitions of particle size to determine the depth of field. Our paper determines the depth of field for a wide range of crystal habits and examines the variability between them depending on the particle size metric used. We find that if circle equivalent diameter is used to calculate the depth of field then there is less variability between habits compared to using the mean X-Y or maximum dimension.

We have added the following paragraph to section 4.1 on the calculation of the effective array width, which should use the image size parallel to the optical array

"When determining the effective array width (Eq. 2), the image size along the direction of the photodiode array should be used. However, this size is a function of the particle's Z position, which is the reason why the effective array width needs to be integrated over the depth of field to determine the sample volume (Eq. 2). This can be calculated using the AST model if the true particle shape can be assumed (e.g. spherical particles in liquid cloud). However, if the true particle shape is not known, as is often the case for ice clouds, then it remains a source of uncertainty in the calculated sample volume."

Third, the paper exaggerates the implications of the study for the parameterizations or representations of small ice crystals. Most previous studies have specifically noted that there are large uncertainties in quantifying the contributions of particles with dimensions smaller than 150 micrometers due to small and poorly defined depths of field for small particles (going back to a study of Baumgardner and Korolev 1997 that has been cited many times). This study is not even referenced here! This study seems to be using a threshold size of 200 um rather than 150 um, so there is a bit of a difference here. But, the findings of the manuscript should be better placed in the context of other studies that have already been conducted as otherwise the implications of this study are overexaggerated.

There are several uncertainties associated with measuring small particles using OAPs: electronic time response; poorly defined depth of field; poor counting statistics due to a small depth of field; and miss-sizing due to diffraction. These have been addressed to varying degrees in the literature.

Baumgardner and Korolev (1997) discusses the uncertainty due to the electronic time response of the probe. This is already referenced in Sect 2.1 of the paper:

"Baumgardner & Korolev (1997) show that the electronic time response of older probes can significantly reduce the DoF of small particles. This affect has been minimised in more modern probes such as the 2D-S and CIP-15, which have an order of magnitude faster time response."

Counting uncertainty can be relatively simply quantified and can be mitigated by using longer averaging times. Generally, previous studies address this source of uncertainty (see for example the recent cirrus climatology paper Kramer et al., 2020).

However, studies examining the sizing uncertainty due to the diffraction from non-spherical shapes are rare (we are only aware of Vaillant de Guélis et al., 2019). No corrections for OAP PSDs of non-spherical particles exist as they do for spherical particles (e.g. Korolev et al., 2007; O'Shea et al., 2019). Despite this uncertainty, OAP measurements of ice crystal PSDs for particles smaller than 150 um are still widely used in parameterisations (e.g. Delanoe et al., 2014) and process studies (e.g. Cotton et al., 2013; Jackson et al., 2015).

I'm not sure that the data availability statement meets the threshold required by the journal. In general, the data should be available at a publicly accessible web site rather than only on request to the contact author. If this study is going to have far-reaching implications, these data should be more openly available for others to test their algorithms with. I'll leave it for the Chief Editor to decide if this statement is adequate.

We will follow the editor's advice regarding this.

References

Baumgardner, D. and Korolev, A.: Airspeed Corrections for Optical Array Probe Sample Volumes, J.

Atmos. Ocean. Tech., 14, 1224–1229, https://doi.org/10.1175/1520-0426(1997)014<1224:ACFOAP>2.0.CO;2, 1997.

Cotton, R. J., Field, P. R., Ulanowski, Z., Kaye, P. H., Hirst, E., Greenaway, R. S., Crawford, I., Crosier, J., and Dorsey, J.: The effective density of small ice particles obtained from in situ aircraft observations of mid-latitude cirrus, Q. J. Royal Meteor. Soc., 139, 1923–1934, 2013.

Davis, S., Hlavka, D., Jensen, E., Rosenlof, K., Yang, Q., Schmidt, S., Borrmann, S., Frey, W., Lawson, P., Voemel, H., and Voemel, T. P.: In situ and lidar observations of tropopause subvisible cirrus clouds during TC4, J. Geophys. Res.-Atmos., 115, D00J17, doi:10.1029/2009JD013093, 2010.

Delanoë, J. M. E., Heymsfield, A. J., Protat, A., Bansemer, A., and Hogan, R. J.: Normalized particle size distribution for remote sensing application, J. Geophys. Res.-Atmos., 119, 4204–4227, https://doi.org/10.1002/2013JD020700, 2014.

Gurganus, C. and Lawson, P.: Laboratory and flight tests of 2D imaging probes: Toward a better understanding of instrument performance and the impact on archived data, J. Atmos. Ocean. Tech., 35, 7, 1533–1553, https://doi.org/10.1175/JTECH-D-17- 0202.1, 2018.

Jackson, R. C., McFarquhar, G. M., Fridlind, A. M., and Atlas, R.: The dependence of cirrus gamma size distributions expressed as volumes in N0-λ-μ phase space and bulk cloud properties on environmental conditions: Results from the Small Ice Particles in Cirrus Experiment (SPARTICUS), J. Geophys. Res.-Atmos., 120, 10351–10377, doi:10.1002/2015JD023492, 2015.

Korolev, A.: Reconstruction of the sizes of spherical particles from their shadow images. Part I: Theoretical considerations, J. Atmos. Ocean. Tech., 24, 376–389, 2007.

Krämer, M., Rolf, C., Spelten, N., Afchine, A., Fahey, D., Jensen, E., Khaykin, S., Kuhn, T., Lawson, P., Lykov, A., Pan, L. L., Riese, M., Rollins, A., Stroh, F., Thornberry, T., Wolf, V., Woods, S., Spichtinger, P., Quaas, J., and Sourdeval, O.: A Microphysics Guide to Cirrus – Part II: Climatologies of Clouds and Humidity from Observations, Atmos. Chem. Phys. Discuss., https://doi.org/10.5194/acp-2020-40, in review, 2020.

McFarquhar, G. M., Baumgardner, D., Bansemer, A., Abel, S. J., Crosier, J., French, J., Rosenberg, P., Korolev, A., Schwarzoen- boeck, A., Leroy, D., Um, J., Wu, W., Heymsfield, A. J., Twohy, C., Detwiler, A., Field, P., Neumann, A., Cotton, R., Axisa, D., and Dong, J.: 11 – Processing of Ice Cloud In Situ Data Collected by Bulk Water, Scattering, and Imaging Probes: Fundamentals, Uncertainties, and Efforts toward Consistency, Meteor. Mon., 58, 11.1–11.33, https://doi.org/10.1175/AMSMONOGRAPHS- D-16-0007.1, 2017

O'Shea, S. J., Crosier, J., Dorsey, J., Schledewitz, W., Crawford, I., Borrmann, S., Cotton, R., and Bansemer, A.: Revisiting particle sizing using greyscale optical array probes: evaluation using laboratory experiments and synthetic data, Atmos. Meas. Tech., 12, 3067–3079, https://doi.org/10.5194/amt-12-3067-2019, 2019.

Vaillant de Guélis, T., Schwarzenböck, A., Shcherbakov, V., Gourbeyre, C., Laurent, B., Dupuy, R., Coutris, P., and Duroure, C.: Study of the diffraction pattern of cloud particles and the respective responses of optical array probes, Atmos. Meas. Tech., 12, 2513–2529, https://doi.org/10.5194/amt-12-2513-2019, 2019.

Wu, W. and McFarquhar, G. M.: On the Impacts of Different Definitions of Maximum Dimension for Nonspherical Particles Recorded by 2D Imaging Probes, J. Atmos. Ocean. Tech., 33, 1057–1072, https://doi.org/10.1175/JTECH-D-15-0177.1, 2016.